# Iteratively Learn Diverse Strategies with State Distance Information

**Wei Fu**[1♭], **Weihua Du**[*1], **Jingwei Li**[*1], **Sunli Chen**[1], **Jingzhao Zhang**[12], **Yi Wu**[12♯]
[1] IIIS, Tsinghua University, [2] Shanghai Qi Zhi Institute
♭ fuwth17@gmail.com, ♯jxwuyi@gmail.com

## Abstract

In complex reinforcement learning (RL) problems, policies with similar rewards may have substantially different behaviors. It remains a fundamental challenge to optimize rewards while also discovering as many *diverse* strategies as possible, which can be crucial in many practical applications. Our study examines two design choices for tackling this challenge, i.e., *diversity measure* and *computation framework*. First, we find that with existing diversity measures, visually indistinguishable policies can still yield high diversity scores. To accurately capture the behavioral difference, we propose to incorporate the state-space distance information into the diversity measure. In addition, we examine two common computation frameworks for this problem, i.e., population-based training (PBT) and iterative learning (ITR). We show that although PBT is the precise problem formulation, ITR can achieve comparable diversity scores with higher computation efficiency, leading to improved solution quality in practice. Based on our analysis, we further combine ITR with two tractable realizations of the state-distance-based diversity measures and develop a novel diversity-driven RL algorithm, *State-based Intrinsic-reward Policy Optimization* (SIPO), with provable convergence properties. We empirically examine SIPO across three domains from robot locomotion to multi-agent games. In all of our testing environments, SIPO consistently produces strategically diverse and human-interpretable policies that cannot be discovered by existing baselines.

## 1 Introduction

A consensus in deep learning (DL) is that different local optima have similar mappings in the functional space, leading to similar losses to the global optimum [62, 56, 40]. Hence, via stochastic gradient descent (SGD), most DL works only focus on the final performance without considering *which* local optimum SGD discovers. However, in complex reinforcement learning (RL) problems, the policies associated with different local optima can exhibit significantly different behaviors [10, 35, 64]. Thus, it is a fundamental problem for an RL algorithm to not only optimize rewards but also discover as many diverse strategies as possible. A pool of diversified policies can be further leveraged towards a wide range of applications, including the discovery of emergent behaviors [34, 3, 61], generating diverse dialogues [30], designing robust robots [12, 25, 19], and enhancing human-AI collaboration [38, 9, 11].

Obtaining diverse RL strategies requires a quantitative method for measuring the difference (i.e., *diversity*) between two policies. However, how to define such a measure remains an open challenge. Previous studies have proposed various diversity measures, such as comparing the difference between the action distributions generated by policies [60, 38, 74], computing probabilistic distances between the state occupancy of different policies [43], or measuring the mutual information between states and

---

[*]Equal Contribution

policy identities [16]. However, it remains unclear which measure could produce the best empirical performance. Besides, the potential pitfalls of these measures are rarely discussed.

In addition to diversity measures, there are two common computation frameworks for discovering diverse policies, including population-based training (PBT) and iterative learning (ITR). PBT directly solves a constrained optimization problem by learning a collection of policies simultaneously, subject to policy diversity constraints [52, 38, 9]. Although PBT is perhaps the most popular framework in the existing literature, it can be computationally challenging [48] since the number of constraints grows quadratically with the number of policies. The alternative framework is ITR, which iteratively learns a single policy that is sufficiently different from previous policies [43, 74]. ITR is a greedy relaxation of the PBT framework and it largely simplifies the optimization problem in each iteration. However, the performance of the ITR framework has not been theoretically analyzed yet, and it is often believed that ITR can be less efficient due to its sequential nature.

We provide a comprehensive study of the two aforementioned design choices. First, we examine the limitations of existing diversity measures in a few representative scenarios, where two policies outputting very different action distributions can still lead to similar state transitions. In these scenarios, state-occupancy-based measures are not sufficient to truly reflect the underlying behavior differences of the policies either. By contrast, we observe that diversity measures based on *state distances* can accurately capture the visual behavior differences of different policies. Therefore, we suggest that an effective diversity measure should explicitly incorporate state distance information for the best practical use. Furthermore, for the choice of computation framework, we conduct an in-depth analysis of PBT and ITR. We provide theoretical evidence that ITR, which has a simplified optimization process with fewer constraints, can discover solutions with the same reward as PBT while achieving *at least half* of the diversity score. This finding implies that although ITR is a greedy relaxation of PBT, their optimal solutions can indeed have comparable qualities. Furthermore, note that policy optimization is much simplified in ITR, which suggests that ITR can result in much better empirical performances and should be preferred in practice.

Following our insights, we combine ITR and a state-distance-based diversity measure to develop a generic and effective algorithm, *State-based Intrinsic-reward Policy Optimization (SIPO)*, for discovering diverse RL strategies. In each iteration, we further solve this constrained optimization problem via the Lagrange method and two-timescale gradient descent ascent (GDA) [31]. We theoretically prove that our algorithm is guaranteed to converge to a neighbor of $\epsilon$-stationary point. Regarding the diversity measure, we provide two practical realizations, including a straightforward version based on the RBF kernel and a more general learning-based variant using Wasserstein distance.

We evaluate SIPO in three domains ranging from single-agent continuous control to multi-agent games: Humanoid locomotion [42], StarCraft Multi-Agent Challenge [57], and Google Research Football (GRF) [26]. Our findings demonstrate that SIPO surpasses baselines in terms of population diversity score across all three domains. Remarkably, our algorithm can successfully discover 6 distinct human-interpretable strategies in the GRF 3-vs-1 scenario and 4 strategies in two 11-player GRF scenarios, namely counter-attack, and corner, without any domain-specific priors, which are beyond the capabilities of existing algorithms.

## 2 Related Work

**Diversity in RL.** It has been shown that policies trained under the same reward function can exhibit significantly different behaviors [10, 35]. Merely discovering a single high-performing solution may not suffice in various applications [12, 64, 25]. As such, the discovery of a diverse range of policies is a fundamental research problem, garnering attention over many years [44, 13, 28]. Early works are primarily based on multi-objective optimization [45, 55, 39, 47, 54], which assumes a set of reward functions is given in advance. In RL, this is also related to reward shaping [46, 2, 15, 61]. We consider learning diverse policies without any domain knowledge.

**Population-based training (PBT)** is the most popular framework for diverse solutions by jointly learning separate policies. Representative works include evolutionary computation [65, 37, 52], league training [64, 23], computing Hessian matrix [51] or constrained optimization with a population diversity measure [38, 73, 29, 36, 9]. An improvement is to learn a latent variable policy instead of separate ones. Prior works have incorporated different domain knowledge to design the latent code,

such as action clustering [66], agent identities [29, 21] or prosocial level [53, 3]. The latent variable can be also learned in an unsupervised fashion, such as in DIYAN [16] and its variants [25, 49]. Zahavy et al. [71] learns diverse policies with hard constraints on rewards to ensure the derived policies are (nearly) optimal, potentially hindering policies with disparate reward scales. On the other hand, our method prioritizes diversity and fully accepts sub-optimal strategies.

**Iterative learning (ITR)** simplifies PBT by only optimizing a single policy in each iteration and forcing it to behave differently w.r.t. previously learned ones [43, 60, 74]. While some ITR works require an expensive clustering process before each iteration [72] or domain-specific features [70], we consider domain-agnostic ITR in an end-to-end fashion. Besides, Pacchiano et al. [50] learns a kernel-based score function to iteratively guide policy optimization. The score function is conceptually similar to SIPO-WD but is applied to a parallel setting with more restricted expressiveness power.

**Diversity Measure.** Most previous works considered diversity measures on action distribution and state occupancy. For example, measures such as Jensen-Shannon divergence [38] and cross-entropy [74] are defined over policy distributions to encourage different policies to take different actions on the same state, implicitly promoting the generation of diverse trajectories. Other measures such as maximum mean discrepancy [43] maximize the probability distance between the state distributions induced by two policies. However, these approaches can fail to capture meaningful behavior differences between two policies in certain scenarios, as we will discuss in Section 4.1. There also exist specialized measures, such as cross-play rewards [9], which are designed for cooperative multi-agent games. It is worth noting that diversity measures are closely related to exploration criteria [4, 20, 6, 27] and skill discovery [8, 32, 24], where a diversity surrogate objective is often introduced to encourage broad state coverage. However, this paper aims to explicitly discover mutually distinct policies. Our diversity measure depends on a function that computes the distance between states visited by two policies.

## 3 Preliminary

**Notation:** We consider POMDP [59] defined by $\mathcal{M} = \langle \mathcal{S}, \mathcal{A}, \mathcal{O}, r, P, O, \nu, H \rangle$. $\mathcal{S}$ is the state space. $\mathcal{A}$ and $\mathcal{O}$ are the action and observation space. $r : \mathcal{S} \times \mathcal{A} \to \mathbb{R}$ is the reward function. $O : \mathcal{S} \to \mathcal{O}$ is the observation function. $H$ is the horizon. $P$ is the transition function. At timestep $h$, the agent receives an observation $o_h = O(s_h)$ and outputs an action $a_h \in \mathcal{A}$ w.r.t. its policy $\pi : \mathcal{O} \to \triangle(\mathcal{A})$. The RL objective $J(\pi)$ is defined by $J(\pi) = \mathbb{E}_{(s_h, a_h) \sim (P, \pi)} \left[ \sum_{h=1}^{H} r(s_h, a_h) \right]$.

The above formulation can be naturally extended to cooperative multi-agent settings, where $\pi$ and $R$ correspond to the joint policy and the shared reward. We follow the standard POMDP notations for conciseness. Our method will be evaluated in both single-agent tasks and complex cooperative multi-agent scenarios. Among them, multi-agent environments encompass a notably more diverse range of potential winning strategies, and hence offer an apt platform for assessing the effectiveness of our method. Moreover, in this paper, we **assume access to object-centric information and features** rather than pure visual observations to simplify our discussion. We remark that although we restrict the scope of this paper to states, our method can be further extended to high-dimensional inputs (e.g. images, see App. B.4.1) or tabular MDPs via representation learning [68, 14].

Finally, to discover diverse strategies, we aim to learn a set of $M$ policies $\{\pi_i\}_{i=1}^{M}$ such that all of these policies are locally optimal under $J(\cdot)$ but mutually distinct subject to some diversity measure $D(\cdot, \cdot) : \triangle \times \triangle \to \mathbb{R}$, which captures the difference between two policies.

**Existing Diversity Measures:** We say a diversity measure $D$ is defined over action distribution if it can be written as

$$D(\pi_i, \pi_j) = \mathbb{E}_{s \sim q(s)} \left[ \tilde{D}_{\mathcal{A}} \left( \pi_i(\cdot \mid s) \| \pi_j(\cdot \mid s) \right) \right], \tag{1}$$

where $q$ is an occupancy measure over states, $\tilde{D}_{\mathcal{A}} : \triangle \times \triangle \to \mathbb{R}$ measures the difference between action distributions. $\tilde{D}_{\mathcal{A}}$ can be any probability distance as defined in prior works [60, 38, 74, 52].

Denote the state occupancy of $\pi$ as $q_\pi$. We say a diversity measure is defined over state occupancy if it can be written as

$$D(\pi_i, \pi_j) = \tilde{D}_{\mathcal{S}} \left( q_{\pi_i} \| q_{\pi_j} \right), \tag{2}$$

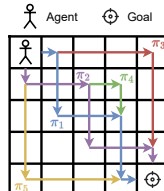

Table 1: Diversity measures of the grid-world example. Computation details can be found in App. B.

| | human | action-based | | | | state-distance-based | |
|---|---|---|---|---|---|---|---|
| | | KL | $\text{JSD}_1$ | $\text{JSD}_0$/EMD | $L_2$ norm | $L_2$ norm | EMD |
| $D(\pi_1, \pi_2)$ | small | $+\infty$ | $\log 2$ | $1/2$ | $\sqrt{7}$ | $2\sqrt{2}$ | $5.7$ |
| $D(\pi_1, \pi_3)$ | large | $+\infty$ | $\log 2$ | $1/8$ | $1$ | $\mathbf{2\sqrt{6}}$ | $\mathbf{11.3}$ |

Figure 1: (left) A grid-world environment with 5 different optimal policies. Intuitively, $D(\pi_1, \pi_2) < D(\pi_1, \pi_3)$ and $D(\pi_3, \pi_4) < D(\pi_3, \pi_5)$. However, action-based measures can give $D_{\mathcal{A}}(\pi_1, \pi_2) \geq D_{\mathcal{A}}(\pi_1, \pi_3)$ and state-occupancy-based measures can give $D(\pi_3, \pi_4) = D(\pi_3, \pi_5)$.

which can be realized as an integral probability metric [43]. We remark that $q_\pi$ is usually intractable.

In addition to diversity measures, we present two popular computation frameworks for this purpose.

**Population-Based Training (PBT):** PBT is a straightforward formulation by jointly learning $M$ policies $\{\pi_i\}_{i=1}^{M}$ subject to pairwise diversity constraints, i.e.,

$$\max_{\{\pi_i\}} \sum_{i=1}^{M} J(\pi_i) \ \text{ s.t. } D(\pi_j, \pi_k) \geq \delta, \forall j, k \in [M], j \neq k, \tag{3}$$

where $\delta$ is a threshold. In our paper, we consistently refer to the aforementioned computation framework as "PBT", rather than adjusting hyperparameters [22]. Despite a precise formulation, PBT poses severe optimization challenges due to mutual constraints.

**Iterative Learning (ITR):** ITR is a greedy approximation of PBT by iteratively learning novel policies. In the $i$-th ($1 \leq i \leq M$) iteration, ITR solves

$$\pi_i^\star = \arg\max_{\pi_i} J(\pi_i) \ \text{ s.t. } D(\pi_i, \pi_j^\star) \geq \delta, \forall 1 \leq j < i. \tag{4}$$

$\pi_j^\star$ is recursively defined by the above equation. Compared with PBT, ITR trades off wall-clock time for less required computation resources (e.g., GPU memory) and performs open-ended training (i.e., the population size $M$ does not need to be fixed at the beginning of training).

## 4 Analysis of Existing Diversity-Discovery Approaches

In this section, we conduct both quantitative and theoretical analyses of existing approaches to motivate our method. We first discuss diversity measures in Sec. 4.1 and then compare computation frameworks, namely PBT and ITR, in Sec. 4.2.

### 4.1 A Common Missing Piece in Diversity Measure: State Distance

The perception of diversity among humans primarily relies on the level of dissimilarity within the state space, which is measured by a distance function. However, the diversity measures outlined in Eq. (1) and Eq. (2) completely fail to account for such crucial information. In this section, we provide a detailed analysis to instantiate this observation with concrete examples and propose a novel diversity measure defined over state distances.

First, we present a synthetic example to demonstrate the limitations of current diversity measures. Our example consists of a grid-world environment with a single agent and grid size $N_G$. The agent starts at the top left of the grid-world and must navigate to the bottom right corner, as shown in Fig. 1. While $N_G$ can be large in general, we illustrate with $N_G = 5$ for simplicity. We draw five distinct policies, denoted as $\pi_1$ through $\pi_5$, which differ in their approach to navigating the grid-world. Consider $\pi_1$, $\pi_2$, and $\pi_3$ first. Although humans may intuitively perceive that policies $\pi_1$ and $\pi_2$, which move along the diagonal, are more similar to each other than to $\pi_3$, which moves along the boundary, diversity measures based on actions can fail to reflect this intuition, as shown in Table 1. Then, let's switch to policies $\pi_3$, $\pi_4$, and $\pi_5$. We find that state-occupancy-based diversity measures are unable to differentiate between $\pi_4$ and $\pi_5$ in contrast to $\pi_3$. This is because the states visited by $\pi_3$ are entirely disjoint from those visited by both $\pi_4$ and $\pi_5$. However, humans would judge $\pi_5$ to be more distinct from $\pi_3$ than $\pi_4$ because both $\pi_3$ and $\pi_4$ tend to visit the upper boundary.

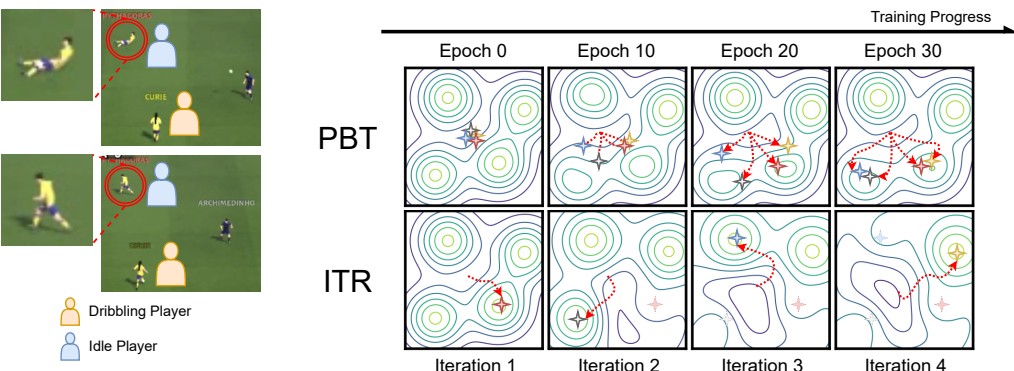

Figure 2: A counter-example for action-based diversity measure: in a football game, we can achieve a high diversity score by simply asking a single idle player to output random actions, which does not affect the high-level gameplay strategy at all.

Figure 3: Illustration of the learning process of PBT and ITR in a 2-D navigation environment with 4 modes. PBT will not uniformly converge to different landmarks as computation can be either too costly or unstable. By contrast, ITR repeatedly excludes a particular landmark, such that policy in the next iteration can continuously explore until a novel landmark is discovered.

Next, we consider a more realistic and complicated multi-agent football scenario, i.e., the Google Research Football [26] environment, in Fig. 2, where an idle player in the backyard takes an arbitrary action without involving in the attack at all. Although the idle player stays still with no effect on the team strategy, action-based measures can produce high diversity scores. This example underscores a notable issue. If action-based measures are leveraged to optimize diversity, the resultant policies can produce visually similar behavior. While it can be possible to exclude idle actions by modifying task rewards, it requires domain-specific hacks and engineering efforts. The issue of idle actions exists even in such popular MARL benchmarks. Similar issues have also been observed in previous works [38].

To summarize, existing measures suffer from a significant limitation — they only compare the behavior trajectories *implicitly* through the lens of action or state distribution without *explicitly measuring state distance*. Specifically, action-based measures fail to capture the behavioral differences that may arise when similar states are reached via different actions. Similarly, state occupancy measures do not quantify *the degree of dissimilarity* between states. To address this limitation, we propose a new diversity measure that explicitly takes into account the distance function in state space:

$$D(\pi_i, \pi_j) = \mathbb{E}_{(s,s')\sim\gamma} \left[ g \left( d \left( s, s' \right) \right) \right], \tag{5}$$

$d$ is a distance metric over $\mathcal{S} \times \mathcal{S}$. $g : \mathbb{R}^+ \to \mathbb{R}$ is a monotonic cost function. $\gamma \in \Gamma(q_{\pi_i}, q_{\pi_j})$ is a distribution over state pairs. $\Gamma(q_{\pi_i}, q_{\pi_j})$ denotes the collection of all distributions on $\mathcal{S} \times \mathcal{S}$ with marginals $q_{\pi_i}$ and $q_{\pi_j}$ on the first and second factors respectively. The cost function $g$ is a notation providing a generalized and unified definition. It also contributes to training stability by scaling the raw distance. We highlight that Eq. (5) computes the cost on individual states before taking expectation, and therefore prevents information loss of taking the average over the entire trajectory (e.g. the DvD score [52]). We also note that states are consequences of performed actions. Hence, a state-distance-based measure also implicitly reflects the (meaningful) differences in actions between two policies. We compute two simple measures based on state distance, i.e., the $L_2$ norm and the Earth Moving Distance (EMD), for the grid-world example and present results in Table 1. These measures are consistent with human intuition.

## 4.2   Computation Framework: Population-Based or Iterative Learning?

We first consider the simplest motivating example to intuitively illustrate the optimization challenges. Let's assume that $\pi_i$ is a scalar, $J(\pi_i)$ is linear in $\pi_i$, and $D(\pi_i, \pi_j) = |\pi_i - \pi_j|$. In our definition, where $M$ denotes the number of diverse policies, PBT involves $\Theta(M^2)$ constraints in a single linear programming problem while ITR involves $\mathcal{O}(M)$ constraints in each of $M$ iterations. Given that the complexity of linear programming is a high-degree polynomial (higher than 2) of the number of constraints, solving PBT is harder (and probably slower) than solving ITR in a total of $M$ iterations,

*despite PBT being parallelized.* This challenge can be more severe in RL due to complex solution space and large training variance.

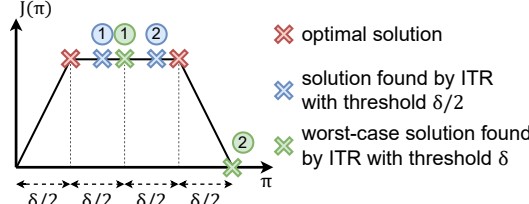

Although ITR can be optimized efficiently, it remains unclear whether ITR, as a greedy approximation of PBT, can obtain solutions of comparable rewards. Fig. 4 shows the worst case in the 1-D setting when the ITR solutions (green) can indeed have lower rewards than the PBT solution (red) subject to the same diversity constraint. However, we will show in the next theorem that ITR is guaranteed to have no worse rewards than PBT by trading off half of the diversity.

Figure 4: 1-D worst case of ITR. With threshold $\delta$, ITR finds solutions with inferior rewards. However, ITR can find optimal solutions if the threshold is halved.

**Theorem 4.1.** *Assume $D$ is a distance metric. Denote the optimal value of Eq.( 3) as $T_1$. Let $T_2 = \sum_{i=1}^{M} J(\tilde{\pi}_i)$ where*

$$\tilde{\pi}_i = \arg\max_{\pi_i} J(\pi_i) \ \ s.t. \ D(\pi_i, \tilde{\pi}_j) \geq \delta/2, \forall 1 \leq j < i \tag{6}$$

*for $i = 1, \ldots, M$, then $T_2 \geq T_1$.*

Please see App. E.1 for the proof. The above theorem provides a quality guarantee for ITR. The proof can be intuitively explained by the 1-D example in Fig. 4, where green points represent the worst case with threshold $\delta$ and blue points represent the solutions with threshold $\delta/2$. Thm. 4.1 shows that, for any policy pool derived by PBT, we can always use ITR to obtain another policy pool, which has *the same rewards* and *comparable diversity scores*.

**Empirical Results:** We empirically compare PBT and ITR in a 2-D navigation environment with 1 agent and $N_L$ landmarks in Fig. 3. The reward is 1 if the agent successfully navigates to a landmark and 0 otherwise. We train $N_L$ policies using both PBT and ITR to discover strategies toward each of these landmarks. More details can be found in App. D. Table 2 shows the number of discovered landmarks by PBT and ITR. ITR performs consistently better than PBT even in this simple example. We intuitively illustrate the learning process of PBT and ITR in Fig. 3. ITR, due to its computation efficiency, can

Table 2: The number of discovered landmarks across 6 seeds with standard deviation in the bracket.

| setting | PBT | ITR |
|---------|-----|-----|
| $N_L = 4$ | 2.0 (1.0) | **3.5** (0.5) |
| $N_L = 5$ | 2.2 (0.9) | **4.5** (0.5) |

afford to run longer iterations and tolerate larger exploration noises. Hence, it can converge easily to diverse solutions by imposing a large diversity constraint. PBT, however, only converges when the exploration is faint, otherwise it diverges or converges too slowly.

### 4.3 Practical Remark

Based on the above analyses, we suggest ITR and diversity measures based on state distances be *preferred* in RL applications. We also acknowledge that, by the no-free-lunch theorem, they cannot be universal solutions and that trade-offs may still exist (see discussions in Sec.7 and App.F). Nonetheless, in the following sections, we will show that the effective implementation of these choices can lead to superior performances in various challenging benchmarks. We hope that our approach will serve as a starting point and provide valuable insights into the development of increasingly powerful algorithms for potentially more challenging scenarios.

## 5 Method

In this section, we develop a diversity-driven RL algorithm, *State-based Intrinsic-reward Policy Optimization (SIPO)*, by combining ITR and state-distance-based measures. SIPO runs $M$ iterations to discover $M$ distinct policies. At the $i$-th iteration, we solve equation (4) by converting it into unconstrained optimization using the Lagrange method. The unconstrained optimization can be written as:

$$\min_{\pi_i} \max_{\lambda_j \geq 0, \ 1 \leq j < i} -J(\pi_i) - \sum_{j=1}^{i-1} \lambda_j \left( D_{\mathcal{S}}(\pi_i, \pi_j^\star) - \delta \right) \tag{7}$$

$\lambda_j$ ($1 \leq j < i$) are Lagrange multipliers. $\{\pi_j^\star\}_{j=1}^{i-1}$ are previously obtained policies. We adopt two-timescale Gradient Descent Ascent (GDA) [31] to solve the above minimax optimization, i.e., performing gradient descent over $\pi_i$ and gradient ascent over $\lambda_j$ with different learning rates. In our algorithm, we additionally enforce the dual variables $\lambda_j$ to be bounded (i.e., in an interval $[0, \Lambda]$ for a large number $\Lambda$), which plays an important role both in the theoretical analysis and in empirical convergence. However, $D_\mathcal{S}(\pi_i, \pi_j^\star)$ cannot be directly optimized w.r.t. $\pi_i$ through gradient-based methods because it depends on the states traversed by $\pi$, rather than its output (e.g. actions). Therefore, we cast $D_\mathcal{S}(\pi_i, \pi_j^\star)$ as the cumulative sum of intrinsic rewards, specifically the intrinsic return. This allows us to leverage policy gradient techniques for optimization. The pseudocode of SIPO can be found in App. G.

An important property of SIPO is the convergence guarantee. We present an informal illustration in Thm. 5.1 and present the formal theorem with proof in App. E.2.

**Theorem 5.1.** *(Informal) Under continuity assumptions, SIPO converges to an $\epsilon$-stationary point.*

**Remark:** We assumed that the return $J$ and the distance $D_\mathcal{S}$ are smooth in policies. In practice, this is true if (1) policy and state space are bounded and (2) reward function and system dynamics are continuous in the policy. (Continuous functions are bounded over compact spaces.) The key step is to analyze the role of the bounded dual variables $\lambda$, which achieves an $\frac{1}{\Lambda}$-approximation of constraint without hurting the optimality condition.

Instead of directly defining $D_\mathcal{S}$, we define intrinsic rewards as illustrated in Sec. 5, such that $D_\mathcal{S}(\pi_i, \pi_j^\star) = \mathbb{E}_{s_h \sim \mu_{\pi_i}} \left[ \sum_{h=1}^{H} r_{\text{int}}(s_h; \pi_i, \pi_j^\star) \right].$

**RBF Kernel:** The most popular realization of Eq. (5) in machine learning is through kernel functions. Herein, we realize Eq. (5) as an RBF kernel on states. Formally, the intrinsic reward is defined by

$$r_{\text{int}}^{\text{RBF}}(s_h; \pi_i, \pi_j^\star) = \frac{1}{H} \mathbb{E}_{s' \sim \mu_{\pi_j^\star}} \left[ -\exp\left( -\frac{\|s_h - s'\|^2}{2\sigma^2} \right) \right] \tag{8}$$

where $\sigma$ is a hyperparameter controlling the variance.

**Wasserstein Distance:** For stronger discrimination power, we can also realize Eq. (5) as $L_2$-Wasserstein distance. According to the dual form [63], we define

$$r_{\text{int}}^{\text{WD}}(s_h; \pi_i, \pi_j^\star) = \frac{1}{H} \sup_{\|f\|_L \leq 1} f(s_h) - \mathbb{E}_{s' \sim \mu_{\pi_j^\star}} \left[ f(s') \right] \tag{9}$$

where $f : \mathcal{S} \to \mathbb{R}$ is a 1-Lipschitz function. This realization holds a distinct advantage due to its interpretation within optimal transport theory [63, 1]. Unlike distances that rely solely on specific summary statistics such as means, Wasserstein distance can effectively quantify shifts in state distributions and remains robust in the presence of outliers [63]. We implement $f$ as a neural network and clip parameters to $[-0.01, 0.01]$ to ensure the Lipschitz constraint. Note that $r_{\text{int}}^{\text{WD}}$ incorporates representation learning by utilizing a learnable scoring function $f$ and is more flexible in practice. We also show in App. B.4 that $r_{\text{int}}^{\text{WD}}$ is robust to different inputs, including states with random noises and RGB images.

We name SIPO with $r_{\text{int}}^{\text{RBF}}$ and $r_{\text{int}}^{\text{WD}}$ ***SIPO-RBF*** and ***SIPO-WD*** respectively.

**Implementation:** To incorporate temporal information, we stack the recent 4 global states to compute intrinsic rewards and normalize the intrinsic rewards to stabilize training. In multi-agent environments, we learn an agent-ID-conditioned policy [17] and share the parameter across all agents. Our implementation is based on MAPPO [69] with more details in App. D.

## 6 Experiments

We evaluate SIPO across three domains that exhibit multi-modality of solutions. The first domain is the humanoid locomotion task in Isaac Gym [42], where diversity can be quantitatively assessed by well-defined behavior descriptors. We remark that the issues we addressed in Sec. 4.1 may not be present in this task where the action space is small and actions are highly correlated with states. Further, we examine the effectiveness of SIPO in two much more challenging multi-agent domains, StarCarft Multi-Agent Challenge (SMAC) [57] and Google Research Football (GRF) [26], where well-defined behavior descriptors are not available and existing diversity measures may produce misleading diversity scores. We provide introductions to these environments in App. C.

First, we show that SIPO can efficiently learn diverse strategies and outperform several baseline methods, including DIPG [43], SMERL [25], DvD [52], and RSPO [74]. Then, we qualitatively demonstrate the emergent behaviors learned by SIPO, which are both *visually distinguishable* and *human-interpretable*. Finally, we perform an ablation study over the building components of SIPO and show that both the diversity measure, ITR, and GDA are critical to the performance.

All algorithms run for the same number of environment frames on a desktop machine with an RTX3090 GPU. Numbers are average values over 5 seeds in Humanoid and SMAC and 3 seeds in GRF with standard deviation shown in brackets. More algorithm details can be found in App. D. Additional visualization results can be found on our project website (see App. A).

## 6.1 Comparison with Baseline Methods

**Humanoid Locomotion.** Following Zhou et al. [74], we train a population of size $4$. We assess diversity by the pairwise distance of joint torques, a widely used behavior descriptor in recent Quality-Diversity works [67]. Torque states are not included as the input of diversity measures and we only use them for evaluation to ensure a fair comparison. Results are shown in Table 3. We can see that both variants of SIPO can outperform all baseline methods except that SIPO-RBF achieves comparable performance with RSPO, even if RSPO explicitly encourages the output of different actions/forces.

Table 3: Pairwise distance of joint torques (i.e., diversity scores) in the humanoid locomotion task.

| SIPO-RBF | SIPO-WD | RSPO |
|---|---|---|
| 0.53(0.17) | **0.71(0.23)** | 0.53(0.05) |

| DIPG | DvD | SMERL |
|---|---|---|
| 0.12(0.04) | 0.40(0.22) | 0.01(0.00) |

**SMAC** Following Zhou et al. [74], we run SIPO and all baselines on an easy map, *2m_vs_1z*, and a hard map, *2c_vs_64zg*, both across 4 iterations. We merge all trajectories produced by the policy collection and incorporate a $k$-nearest-neighbor state entropy estimation [58] to assess diversity. Intuitively, a more diverse population should have a larger state entropy value. We set $k = 12$ following Liu and Abbeel [32] and show results in Table 4. On these maps, two agents are both involved in the attack. Therefore, RSPO, which incorporates an action-based cross-

Table 4: State entropy estimated by $k$-nearest-neighbor in SMAC. ($k = 12$)

| | *2m_vs_1z* | *2c_vs_64zg* |
|---|---|---|
| SIPO-RBF | **0.038(0.002)** | **0.072(0.003)** |
| SIPO-WD | 0.036(0.001) | 0.056(0.003) |
| RSPO | 0.032(0.003) | 0.070(0.001) |
| DIPG | 0.032(0.002) | 0.056(0.004) |
| SMERL | 0.028(0.002) | 0.042(0.002) |
| DvD | 0.030(0.002) | 0.057(0.003) |

entropy measure, can perform well across all baselines. However, SIPO explicitly compares the distance between resulting trajectories and can even outperform RSPO, leading to the most diverse population.

**GRF** We consider three academy scenarios, specifically *3v1*, *counterattack* (*CA*), and *corner*. The GRF environment is more challenging than SMAC due to the large action space, more agents, and the existence of duplicate actions. We determine a population size $M = 4$ by balancing resources and wall-clock time across different baselines. Table 5 compares the number of distinct policies (in terms of ball-passing routes, see App. B.3) discovered in the population. Due to the strong adversarial power of our diversity measures and the application of GDA, SIPO is the most efficient and robust — even in the challenging 11-vs-11 *corner* and *CA* scenario, SIPO can effectively discover different winning strategies in just a few iterations across different seeds. By contrast, baselines suffer from learning instability in these challenging environments and tend to discover policies with slight distinctions. We also calculate the estimated state entropy as we did in SMAC. However, we find that this metric cannot distinguish fine-grained ball-passing behaviors in GRF (check our discussions in App. B).

**Remark:** In GRF experiments, when $M$ is small, even repeated training with different random seeds (PG) is a strong baseline (see Table 5). Hence, the numbers are actually restricted in a small interval (with a lower bound equal to PG results and an upper bound equal to $M = 4$), which makes the improvements by SIPO seemingly less significant. However, achieving clear improvements in these challenging applications remains particularly non-trivial. With a population size $M = 10$, SIPO clearly outperforms baselines by consistently discovering one or more additional strategies.

## 6.2 Qualitative Analysis

For SMAC, we present heatmaps of agent positions in Fig. 5. The heatmaps clearly show that SIPO can consistently learn novel winning strategies to conquer the enemy. Fig. 6 presents the learned

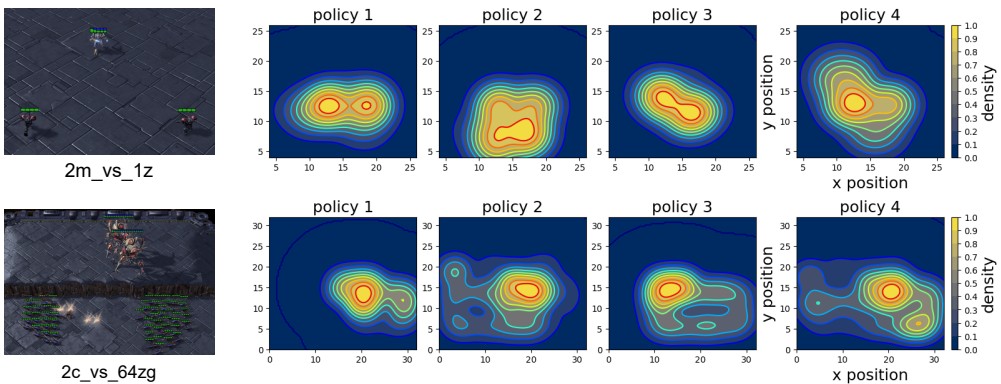

Figure 5: Heatmaps of agent positions in SMAC across 4 iterations with SIPO-RBF.

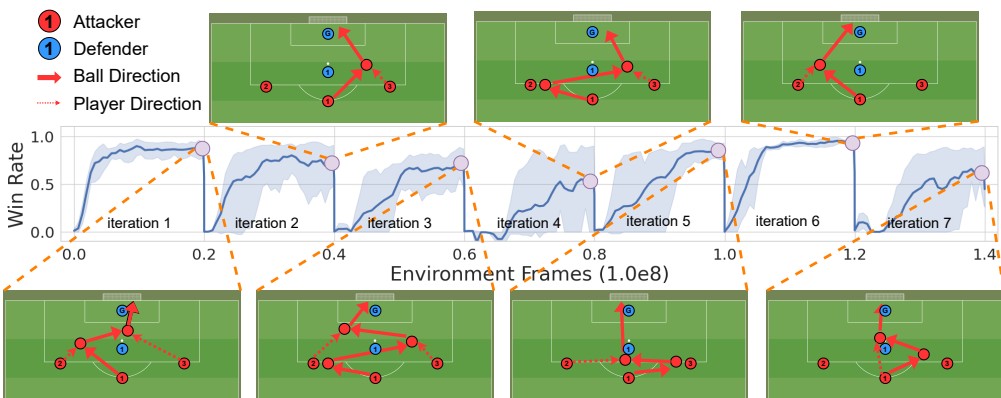

Figure 6: Learning curves and discovered strategies by SIPO-WD in the *3v1* scenario over 7 iterations. Strategies of seed 1 are shown.

behavior by SIPO in the GRF *3v1* scenario of seed 1. We can observe that agents have learned a wide spectrum of collaboration strategies across merely 7 iterations. The strategies discovered by SIPO are both *diverse* and *human-interpretable*. In the first iteration, all agents are involved in the attack such that they can distract the defender and obtain a high win rate. The 2nd and the 6th iterations demonstrate an efficient pass-and-shoot strategy, where agents quickly elude the defender and score a goal. In the 3rd and the 7th iterations, agents learn smart "one-two" strategies to bypass the defender, a prevalent tactic employed by human football players. We note that *NONE* of the baselines have ever discovered this strategy across all runs, while SIPO is consistently able to derive such strategies for all random seeds. Visualization results in *CA* and *corner* scenarios can be found in App. B.

## 6.3  Ablation Study

We apply these changes to SIPO-WD:

- *fix-L*: Fixing the multiplier $\lambda_i$ instead of applying GDA.

Table 5: Number of distinct strategies in GRF discovered by different methods in terms of the ball-passing route. Details of the evaluation protocol can be found in App. B.3.

| | Population Size $M$ | ours | | baselines | | | | random |
|---|---|---|---|---|---|---|---|---|
| | | SIPO-RBF | SIPO-WD | DIPG | SMERL | DvD[1] | RSPO | PG |
| *3v1* | 4 | **3.0 (0.8)** | **3.0 (0.0)** | 2.7 (0.5) | 1.3 (0.5) | **3.0 (0.8)** | 2.0 (0.0) | 2.7 (0.5) |
| *CA* | 4 | **3.3 (0.5)** | 3.0 (0.8) | 2.3 (0.5) | 1.3 (0.5) | - | 2.0 (0.0) | 1.7 (0.5) |
| *corner* | 4 | 2.7 (0.5) | **3.0 (0.8)** | 1.7 (0.5) | 1.0 (0.0) | - | 1.6 (0.5) | 2.0 (0.8) |
| *3v1* | 10 | 4.3 (0.5) | **5.7 (0.5)** | 3.7 (0.5) | - | - | 2.3 (0.5) | - |

[1] Training DvD in *CA* and *corner* or with $M = 10$ requires >24GB GPU memory, which exceeds our memory limit.

- *CE*: The intrinsic reward is replaced with cross-entropy, i.e., $r_{\text{int}}^{\text{CE}}(s_h, a_h) = -\log \pi_j^\star(a_h \mid s_h)$, where $\pi_j^\star$ denotes a previously discovered policy. Additionally, GDA is still applied.

- *filter*: Optimizing the extrinsic rewards on trajectories that have intrinsic returns exceeding $\delta$ and optimizing intrinsic rewards defined by Eq. (9) for other trajectories [74].

- *PBT*: Simultaneously training $M$ policies with $M(M-1)/2$ constraints (i.e., directly solving Eq. (3)) with intrinsic rewards defined by Eq. (9) and GDA.

We report the number of visually distinct policies discovered by these methods in Table 6. Comparison between SIPO and CE demonstrates that the action-based cross-entropy measure may suffer from duplicate actions in GRF and produce nearly identical behavior by overly exploiting duplicate actions, especially in the *CA* and *corner* scenarios with 11

Table 6: # distinct strategies of ablations in GRF.

|        | ours       | fix-L    | CE        | filter    | PBT       |
|--------|------------|----------|-----------|-----------|-----------|
| *3v1*  | **3.0 (0.0)** | 1.0 (0.0) | 2.7 (0.5) | 1.3 (0.5) | 2.7 (0.5) |
| *CA*   | **3.0 (0.8)** | -[1]     | 2.3 (0.8) | 1.0 (0.0) | -[2]      |
| *corner* | **3.0 (0.8)** | -[1]   | 1.7 (0.5) | 1.0 (0.0) | -[2]      |

[1] Not converged.
[2] Training requires >24GB memory and exceeds our memory limit.

agents. Besides, the fixed Lagrange coefficient, the filtering-based method, and PBT are all detrimental to our algorithm. These methods also suffer from significant training instability. Overall, the state-distance-based diversity measure, ITR, and GDA are all critical to the performance of SIPO.

## 7 Conclusion

We tackle the problem of discovering diverse high-reward policies in RL. First, we demonstrate concrete failure cases of existing diversity measures and propose a novel measure that explicitly compares the distance in state space. Next, we present a thorough comparison between PBT and ITR and show that ITR is much easier to optimize and can derive solutions with comparable quality to PBT. Motivated by these insights, we combine ITR with a state-distance-based diversity measure to develop SIPO, which has provable convergence and can efficiently discover a wide spectrum of human-interpretable strategies in a wide range of environments.

**Limitations:** First, we assume direct access to an object-centric state representation. When such a representation is not available (e.g., image-based observations), representation learning becomes necessary and algorithm performance can be affected by the quality of the learned representations. Second, because ITR requires sequential training, the wall clock time of SIPO can be longer than the PBT alternatives when fixing the total number of training samples. The acceleration of ITR remains an open challenge.

**Future Directions:** Besides addressing the above limitations, we suggest three additional future directions based on our paper. First, a consensus on the best algorithmic formulation of distinct solutions in RL remains elusive. It is imperative to understand diversity in a more theoretical manner. Second, while this paper focuses on single-agent and cooperative multi-agent domains, extending SIPO to multi-agent competitive games holds great potential. Finally, although SIPO/ITR enables open-ended training, it is worth studying how to determine the optimal population size to better balance resources and the diversity of the resulting population.

## Acknowledgement

This project is partially supported by 2030 Innovation Megaprojects of China (Programme on New Generation Artificial Intelligence) Grant No. 2021AAA0150000.

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

# A   Project Website

Check `https://sites.google.com/view/diversity-sipo` for GIF demonstrations.

# B   Additional Results

## B.1   More Qualitative Results

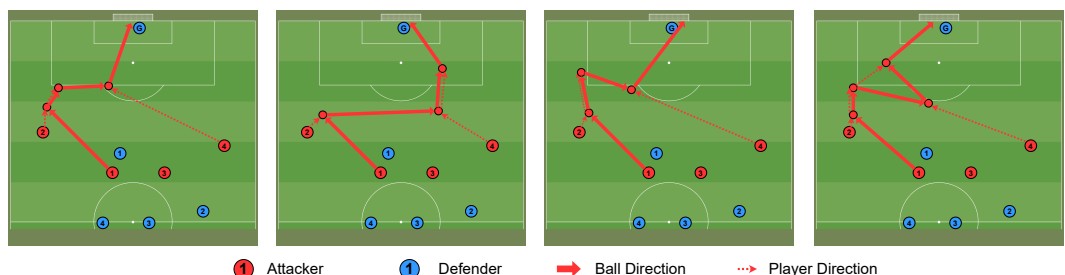

Figure 7: Visualization of learned behaviors in GRF *CA* across a single training trial.

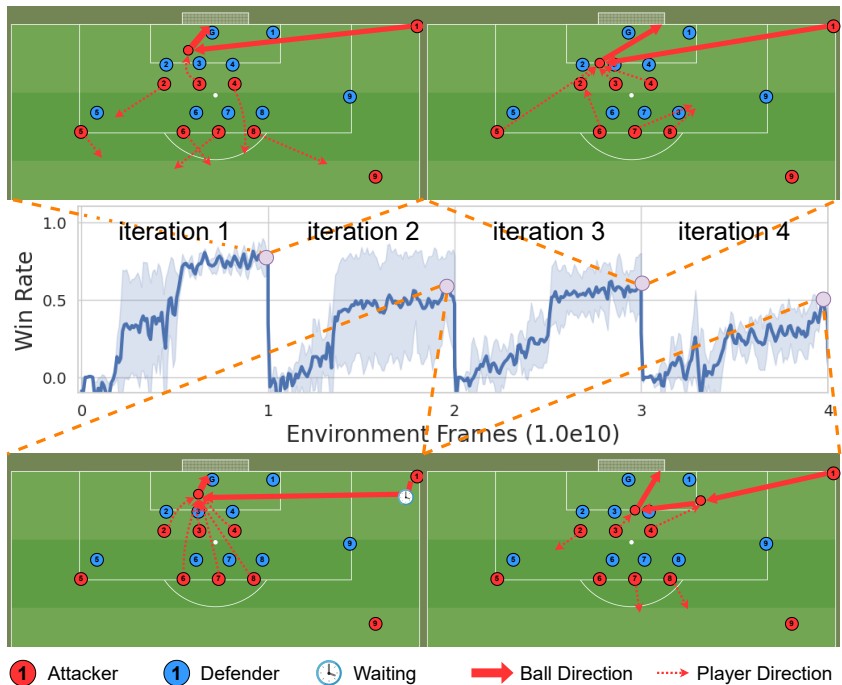

Figure 8: Visualization of learned behaviors in GRF *corner*.

We show additional visualization results in Fig. 7, Fig. 8, and Fig. 9. Corresponding GIF visualizations can be found on our project website.

## B.2   Task Performance Evaluation

The evaluation win rates of the demonstrated visualization results in SMAC and GRF are shown in Table 8. Evaluated episode returns in Humanoid are shown in Table 9.

We also present the diversity score and average rewards achieved by baselines in Table 10. These numerical values are averaged across the entire population for a clear comparison. The tabulated data highlights the varying trade-offs between task performance and diversity exhibited by different

Table 7: $k$-nearest neighbor state entropy estimation in GRF. Population size $M = 4$.

| | ours | | baselines | | | | |
|---|---|---|---|---|---|---|---|
| | SIPO-RBF | SIPO-WD | DIPG | SMERL[1] | DvD[2] | RSPO[1] | PG (random seeding) |
| *3v1* | 0.009(0.000) | 0.012(0.000) | 0.010(0.001) | 0.011(0.002) | 0.010(0.000) | 0.011(0.001) | 0.009(0.001) |
| *CA* | 0.037(0.000) | 0.031(0.006) | 0.036(0.002) | - | - | 0.034(0.001) | 0.039(0.001) |
| *Corner* | 0.028(0.001) | 0.031(0.001) | 0.030(0.002) | - | - | - | 0.028(0.002) |

[1] The learned policy in some iterations cannot even collect a single winning trajectory, so we are unable to compute their diversity score.

[2] Training DvD in *CA* and *corner* requires >24GB GPU memory, which exceeds our memory limit.

algorithms. It is noteworthy that SIPO, in particular, displays an adeptness at training a notably more diverse population while upholding a reasonably moderate level of task performance.

Table 8: Evaluation win rate (%) of the demonstrated visualization results in SMAC and GRF.

| | SMAC | | GRF | | |
|---|---|---|---|---|---|
| | *2m1z* | *2c64zg* | *3v1* | *CA* | *corner* |
| $\pi_1$ | 100.0(0.0) | 98.1(2.1) | 92.3(6.2) | 48.2(10.4) | 78.2(16.2) |
| $\pi_2$ | 99.6(0.9) | 100.0(0.0) | 82.1(8.4) | 43.8(42.2) | 57.0(37.7) |
| $\pi_3$ | 100.0(0.0) | 96.9(3.3) | 90.7(1.1) | 54.7(30.6) | 55.7(20.8) |
| $\pi_4$ | 99.6(0.6) | 98.6(2.4) | 63.6(45.0) | 17.2(30.0) | 30.7(29.0) |
| $\pi_5$ | - | - | 85.4(9.1) | - | - |
| $\pi_6$ | - | - | 93.2(1.9) | - | - |
| $\pi_7$ | - | - | 64.6(32.5) | - | - |

## B.3 Evaluation Metric and Protocol for Diversity

### B.3.1 Humanoid

The Humanoid locomotion task is well-studied in the Quality-Diversity (QD) community, enabling the application of well-defined behavior descriptors (BD) to assess diversity scores. While domain-agnostic metrics like DvD scores can also be applied, we consider domain-specific BDs to be more appropriate and accurate for evaluation in this setting.

### B.3.2 SMAC

Complex multi-agent tasks like SMAC lack well-defined BDs. Hence, domain-agnostic diversity measures such as the state-entropy measure should be applied. Moreover, different SMAC winning strategies tend to visit different areas of the map, which can be usually captured by the state-entropy measure.

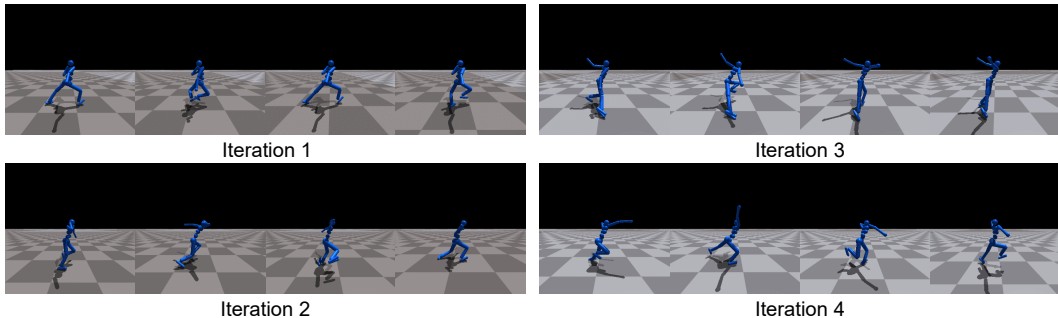

Iteration 1     Iteration 3

Iteration 2     Iteration 4

Figure 9: Visualization of learned behaviors in Humanoid.

Table 9: Episode returns in Humanoid.

| | SIPO-RBF | SIPO-WD | SIPO-WD (visual) |
|---|---|---|---|
| $\pi_1$ | 4863.9(970.3) | 3909.4(533.4) | 4761.3(107.8) |
| $\pi_2$ | 3746.5(488.0) | 3784.2(481.2) | 4349.3(169.0) |
| $\pi_3$ | 3092.0(805.0) | 3770.4(674.4) | 4724.3(946.5) |
| $\pi_4$ | 2332.8(519.8) | 3589.6(387.4) | 3819.7(588.7) |

Table 10: Reward/diversity of all baselines. The reward metric in SMAC and GRF are evaluation win rate (%). The evaluation metrics of diversity used in humanoid, SMAC, GRF are the joint torque distance, state entropy (1e-3), and the number of different ball-passing routes, respectively. It is noteworthy that SIPO, in particular, displays an adeptness at training a notably more diverse population while upholding a reasonably moderate level of task performance.

| Task/Scenario | SIPO-RBF | SIPO-WD | DIPG | RSPO | SMERL | DvD | PPO |
|---|---|---|---|---|---|---|---|
| humanoid | 3508 / 0.53 | 3763 / 0.71 | 5191 / 0.12 | 1455 / 0.53 | 4253 / 0.01 | 4498 / 0.40 | 5299 / - |
| SMAC 2m1z | 100 / 38 | 100 / 36 | 100 / 32 | 100 / 32 | 100 / 28 | 100 / 30 | 100 / - |
| SMAC 2c64zg | 99 / 72 | 93 / 56 | 99 / 70 | 85 / 56 | 100 / 42 | 100 / 57 | 100 / - |
| GRF 3v1 (first 4) | 93 / 3.0 | 82 / 3.0 | 93 / 2.7 | 94 / 2.0 | 91 / 1.3 | 83 / 3.0 | 92 / 2.7 |
| GRF CA | 70 / 3.3 | 41 / 3.0 | 46 / 2.3 | 76 / 2.0 | 45 / 1.3 | - | 50 / 1.7 |
| GRF Corner | 72 / 2.7 | 56 / 3.0 | 75 / 1.7 | 23 / 1.6 | 67 / 1.0 | - | 71 / 2.0 |

### B.3.3 GRF

In our initial study of the GRF task, diversity was evaluated using the $k$-nearest-neighbor state entropy estimation as in SMAC (see Table 7). However, we observed a significant difference between the computed scores and visualized behaviors. Further investigation revealed that state entropy can sometimes report fake diversity in GRF. For example, the ball-moving route is highly fine-grained between nearby players in the counter-attack (CA) scenario, and additional passes may not change the state entropy significantly. Instead, agents' positions play a crucial role in this scenario, where different shooting positions can introduce substantial state variance and lead to a higher entropy score. As an example, readers can refer to the replays of SIPO-RBF (4 iterations of seed 2) and PG (seed 2, 1002, 2002, and 3002), where SIPO-RBF discovers four distinct passing strategies, while PG keeps passing the ball to the same player. Nevertheless, the state entropy of PG (0.0397) is higher than that of SIPO-RBF (0.0378).

Hence, we counted the number of distinct policies according to their ball-passing routes, such as passing the ball to different players or shooting with different players, to evaluate diversity in GRF. To quantify these differences, we extracted the positions of the ball and the players in the field and calculated the nearest ally player ID to the ball across a winning episode. We then removed timesteps where the nearest distance was above a pre-defined threshold of 0.03. Typically, these timesteps correspond to instances when the ball is being transferred among players, making the nearest player ID irrelevant. Next, we removed consecutive duplicate player IDs from the resulting sequence to obtain a concise and informative embedding of the ball passing route. By comparing the lengths of their respective embeddings and verifying that the player IDs in each embedding are identical, we determined whether the two policies exhibit similar behavior.

We acknowledge that existing diversity measures may not be applicable in GRF, and hence we opted for this novel approach to evaluate diversity. Additionally, we experimented with using raw observations, which include ball ownership information provided by the game engine, but found it to be highly inaccurate based on our visualization.

### B.4 Additional Ablation Studies

### B.4.1 Input to the Diversity Measure

**Vectorized States in Google Research Football** We perform an additional ablation study over the input of our diversity measure in GRF *3v1* scenario with SIPO-WD. We consider the following kinds of state input besides the default state input we adopted in Sec. 6:

- full observation (named *full*, 115 dims);
- default state input with random noises of the same dimension (named *random*, 36 dims).

The numbers of visually distinct strategies are listed in Table 11. The performance of *full* and *random* is similarly good. The result implies that the learnable discriminator can automatically filter out irrelevant states to some extent, and that SIPO-WD performs relatively robust w.r.t. different state input of the diversity measure.

Table 11: State input ablation. The table shows the number of distinct strategies in GRF *3v1*.

|  | SIPO-WD | full | random |
|---|---|---|---|
| *3v1* | 3.0 (0.0) | 3.0 (0.8) | 3.0 (0.0) |

**RGB Images in Locomotion Tasks**    We run SIPO-WD in the visual Humanoid task based on Isaac Gym [42]. The training protocol is similar to the state-only version (i.e., the input of policy and intrinsic rewards are both locomotion states of the Humanoid) except that we stack recent 4 RGB camera observations ($84 \times 84$) as the input of intrinsic rewards in Eq. 9. We adopt the training code developed in Isaac Gym and the default PPO configuration. The backbone of the discriminator is composed of 4 convolutional layers with kernel size 3, stride 2, padding 1, and [16, 32, 32, 32] channels. Then the feature is passed to an MLP with 1 hidden layer and 256 hidden units. The activation function is leaky ReLU with slope $0.2$. We also compute the pairwise distance of joint torques as in the state-only version and show the result in Table 12. Visualizations are shown in Fig. 10. SIPO-WD can also learn meaningful diverse behaviors with RGB images as the state input thanks to the learnable Wasserstein discriminator. This implies that our algorithm can be naturally extended to high-dimensional states and incorporated with advances in representation learning, which may be a potential future direction.

### B.4.2   Combining State- and Action-based Diversity Measures

Based on SIPO-RBF, we introduce additional action information by directly concatenating the global state, used for diversity calculation, with the one-hot encoded actions of all agents within the GRF domain. Table 13 presents the outcomes, indicating the number of policies obtained. For scenarios with a limited number of agents, the action-augmented variant demonstrates comparable performance. However, when the agent count increases (as evident in the 11-agent cases of CA and corner), the incorporation of actions can introduce misleading diversity, detracting from the authenticity of the outcomes.

### B.5   How to Adjust Constraint-Related Hyperparameters

Three hyperparameters are essential in the implementation of the intrinsic reward $r_{\text{int}}$: the threshold $\delta$, the intrinsic reward scale factor $\alpha$, and the variance factor $\sigma$ in $r_{\text{int}}^{\text{RBF}}$. These parameters differ under different domains and must be adjusted individually. We find proper parameters by running two

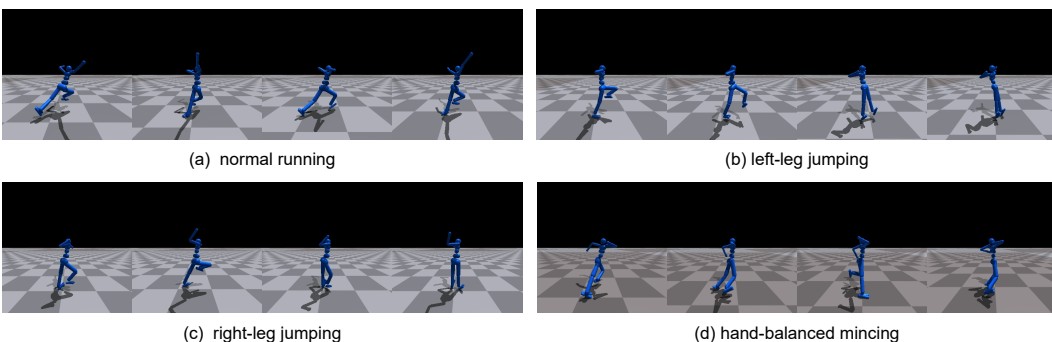

(a) normal running

(b) left-leg jumping

(c) right-leg jumping

(d) hand-balanced mincing

Figure 10: Results of SIPO-WD in the visual Humanoid task.

Table 12: Pairwise distance of joint torques (i.e., diversity score) in Humanoid with visual input. Results in visual experiments are averaged over 3 seeds.

| SIPO-WD (visual) | SIPO-WD | RSPO (best baseline) |
|---|---|---|
| 0.62 (0.26) | 0.71 (0.23) | 0.53 (0.05) |

| | 3v1 | CA | Corner |
|---|---|---|---|
| SIPO-RBF | 3.0 (0.8) | 3.3 (0.5) | 2.7 (0.5) |
| SIPO-RBF w. Action | 3.0 (0.0) | 2.3 (0.5) | 1.0 (0.0) |

Table 13: Ablation study of combining state- and -action-based diversity measures. The number of different strategies across a population of 4 is shown with standard deviation in the brackets.

iterations without constraints and get two similar policies $\pi_0$ and $\pi_1$. We record $r_{\text{int}}$ during training $\pi_1$ and the trend is shown in Fig. 11. Not surprisingly, $r_{\text{int}}$ gradually decreases as training proceeds.

**Threshold** We set $\delta = c_1 D_{\mathcal{S}}(\pi_0, \pi_1)$. We try several different $c_1 \in \{1, 1.2, 1.4, 1.6, 1.8, 2.0\}$ and find that $c_1 = 1.2$ or $1.4$ are universal proper solutions for all the experimental environments.

**Intrinsic Scale Factor** We need to balance the intrinsic reward $r_{int}$ and the original reward $J$ so that neither of the two rewards can dominate the training process. Empirically, the maximums of the two rewards should be in the same order of magnitude. i.e., $\max_\pi J(\pi) = \alpha \times c_2 \lambda_{max} \delta$, where $c_2 = O(1)$. When $c_2$ is too large, the new-trained policy $\pi_j$ will oscillate near the boundary of $D(\pi_i, \pi_j) = \delta$ for some pre-trained policy $p_i$. Conversely, when $c_2$ is too small, the intrinsic reward $r_{int}$ cannot yield diverse strategies. In experiments, we set $c_2 = 1.0$.

**Variance Factor** We sweep the variance factor across $\{1e-3, 5e-3, 1e-2, 2e-2, 1e-3\}$ by training $\pi_1$ and observe the trend of intrinsic rewards. We find the steepest trend and select the corresponding $\sigma$. Empirically, we find that our algorithm performs robustly well when $\sigma^2 = 0.02$.

The $\delta$ and $\alpha$ of GRF and SMAC are listed in Table 14.

## B.6 Computation of Action-Based Measures in the Grid-World Example

We consider the policies illustrated in Fig. 12. These policies are all optimal since these actions only include "right" and "down" and actions on non-visited states can be arbitrary. We only mark actions on states visited by any of these 3 policies and actions on other states can be considered the same.

### B.6.1 Action-Distribution-Based Measures

Action-distribution-based diversity measures can be defined as

$$D_{\mathcal{A}}(\pi_i, \pi_j) = \mathbb{E}_{s \sim q(s)} \left[ \tilde{D} \left( \pi_i(\cdot \mid s) \| \pi_j(\cdot \mid s) \right) \right], \tag{10}$$

where $\tilde{D}(\cdot, \cdot) : \triangle \times \triangle \to \mathbb{R}$ is a measure over action distributions and $q : \triangle(\mathcal{S})$ is a state distribution. Here, we consider $q$ to be the joint state distribution visited by $\pi_i$ and $\pi_j$.

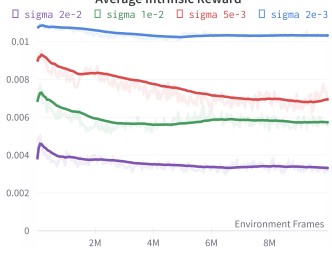

Figure 11: Average intrinsic reward during training $\pi_1$.

Table 14: The values of $\delta$ and $\alpha$ in different environments.

| | football | | | smac | |
|---|---|---|---|---|---|
| | *3v1* | *corner* | *CA* | *2m_vs_1z* | *2c_vs_64zg* |
| $\delta^{\text{WD}}$ | 0.004 | 0.01 | 0.012 | 0.02 | 0.2 |
| $\alpha^{\text{WD}}$ | 1 | 1 | 0.5 | 0.5 | 0.05 |
| $\delta^{\text{RBF}}$ | 0.03 | 0.01 | 0.015 | 0.002 | 0.001 |
| $\alpha^{\text{RBF}}$ | 0.001 | 0.001 | 0.001 | 0.001 | 0.001 |
| $\sigma^2$ | 0.02 | 0.02 | 0.02 | 0.02 | 0.02 |

**KL Divergence** KL divergence is defined by

$$D_{\text{KL}}\left(\pi_i(\cdot \mid s), \pi_j(\cdot \mid s)\right) = \int_{\mathcal{A}} \pi_i(a \mid s) \log \frac{\pi_i(a \mid s)}{\pi_j(a \mid s)} da.$$

When $\pi_j(a \mid s) = 0$ at any state $s$, KL divergence is $+\infty$. Since the trajectories of these policies have disjoint states, $D_{\mathcal{A}}^{\text{KL}}(\pi_1, \pi_2) = D_{\mathcal{A}}^{\text{KL}}(\pi_1, \pi_3) = +\infty$. Similar results can be obtained for cross-entropy.

**JSD$_\gamma$** JSD$_\gamma$ was defined in [38] and we consider two special cases when $\gamma = 0$ and $\gamma = 1$.

As illustrated by [38], JSD$_0$ measures the expected number of times two policies will "disagree" by selecting different actions. On trajectories induced by $\pi_1$ and $\pi_2$, there are $4 + 4$ states that $\pi_1$ disagrees with $\pi_2$ ($\pi_1$ and $\pi_2$ are symmetric) and $D_{\mathcal{A}}^{\text{JSD}_0}(\pi_1, \pi_2) = 8/16 = 1/2$. Similarly, $\pi_1$ and $\pi_3$ only disagree at the initial state, therefore we have $D_{\mathcal{A}}^{\text{JSD}_0}(\pi_1, \pi_3) = 2/16 = 1/8$.

JSD$_1$ is defined by

$$\text{JSD}_1(\pi_i, \pi_j) = -\frac{1}{2} \sum_{\tau_i} P(\tau_i \mid \pi_i) \sum_{t=1}^{T} \frac{1}{T} \log \frac{\pi_i(\tau_i) + \pi_j(\tau_i)}{2\pi_i(\tau_i)}$$

$$-\frac{1}{2} \sum_{\tau_j} P(\tau_j \mid \pi_j) \sum_{t=1}^{T} \frac{1}{T} \log \frac{\pi_i(\tau_j) + \pi_j(\tau_j)}{2\pi_j(\tau_j)}.$$

Since each of the policies considered only induces a single trajectory and $\pi_i(\tau_j) = 0$ ($i \neq j$), we can easily compute

$$D_{\mathcal{A}}^{\text{JSD}_1}(\pi_1, \pi_2) = D_{\mathcal{A}}^{\text{JSD}_1}(\pi_1, \pi_3) = \log 2$$

**Wasserstein Distance** Wasserstein distance or Earth Moving Distance (EMD) is 1 if two policies disagree on a state and 0 otherwise. Therefore, it equals to $D_{\mathcal{A}}^{\text{JSD}_0}$.

### B.6.2 Action Norm

We embed the action "right" as vector $[1, 0]$ since it increases the x-coordinate by 1 and the action "down" as vector $[0, -1]$ since it decreases the y-coordinate by 1. This embedding can be naturally extended to a continuous action space with velocity actions. Following [52], we compute the action norm over a uniform distribution on states. We can see that there are 7 states where $\pi_1$ and $\pi_2$ perform differently and 1 state (the initial state) where $\pi_1$ and $\pi_3$ perform differently. Therefore, we can get $D(\pi_1, \pi_2) = \sqrt{7}$ and $D(\pi_1, \pi_3) = 1$.

### B.6.3 State-Distance-Based Measures

**State $L_2$ Norm** Similar to action $L_2$ norm, we concatenate the coordinates instead of actions as the embedding and compute the $L_2$ norm between embedding.

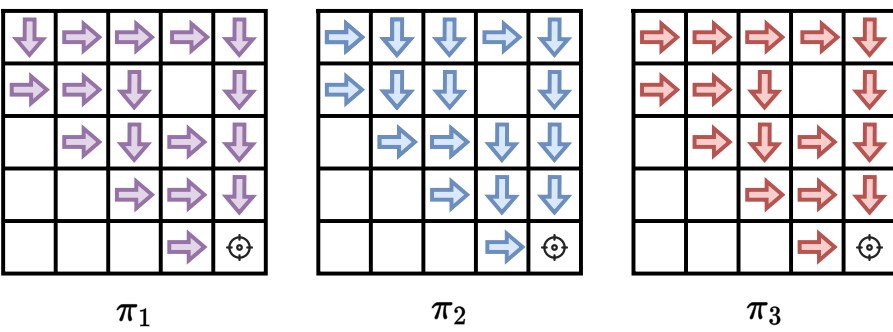

Figure 12: Policies in the grid-world example when $N_G = 5$.

**Wasserstein Distance**   Wasserstein distance is tractable in the grid-world example. We consider 7 states (except the initial and final states) in each trajectory and compute the pair-wise distance as matrix $C$. Then we solve the following linear programming

$$\min_{\gamma} \quad \sum_{i,j} \gamma \odot C$$
$$\text{s.t.} \quad \gamma \mathbf{1} = a, \ \gamma^T \mathbf{1} = b$$
$$\gamma_{i,j} \geq 0$$

where $\odot$ means element-wise multiplication, $\mathbf{1}$ is a all-one vector, $a = [\mathbf{1}^T, \mathbf{0}^T]^T$ and $b = [\mathbf{0}^T, \mathbf{1}^T]^T$ is the marginal state distribution of each policy.

## C   Environment Details

### C.1   Details of the 2D Navigation Environment

The navigation environment has an agent circle with size $a$ and 4 landmark circles with size $b$. We pre-specify a threshold $c$ and constrain that the distance of final states reaching different landmarks must be larger than $c$. Correspondingly, landmark circles are randomly initialized by constraining the pairwise distance between centers to be larger than a threshold $c + 2(a + b)$ such that the final-state constraint is valid. An episode ends if the agent touches any landmarks, i.e., the distance between the center of the agent and the center of the landmark $d < a + b$, or 1000 timesteps have elapsed. The observation space includes the positions of the agent and all landmarks, which is a 10-dimensional vector. The action space is a 2-dimensional vector, which is the agent velocity. The time interval is set to be $\Delta t = 0.1$, i.e., the next position is computed by $x_{t+1} = x_t + \Delta t \cdot v$. The reward is 1 if the agent touches the landmark and 0 otherwise.

### C.2   Details of Environments

We provide training configurations and environment introductions below and refer readers to our project website in App. A for visualizations of these environments.

**Humanoid**   We use the Humanoid environment in IsaacGym [42] with default observation and action spaces. The input of intrinsic rewards or diversity measure is the observation without all torque states.

**SMAC**   We adopt the SMAC environment in the MAPPO codebase[2] with the same configuration as Yu et al. [69]. The input of intrinsic rewards or diversity measure is the state of all allies, including positions, health, etc.

On the "easy" map *2m_vs_1z*, two marines must be controlled to defeat a Zealot. The marines can attack from a distance, while the Zealot's attacks are limited to close range. A successful strategy involves alternating the marines' attacks to distract the Zealot. On the "hard" map *2c_vs_64zg*, two colossi must be controlled by the agents to fight against 64 zergs. The colossi have a wider attack range and can move over cliffs. Strategies on this map may include hit-and-run tactics, waiting in corners, or dividing and conquering enemies. The level of difficulty is determined by the learning performance of existing MARL algorithms. Harder maps require more exploration and training steps.

**GRF**   We adopt the "simple115v2" representation as observation with both "scoring" and "checkpoint" reward. The reward is shared across all agents. The input of intrinsic rewards or diversity measure is the position and velocity of all attackers and the ball. All policies are trained to control the left team to score against built-in bots.

*academy_3_vs_1_with_keeper*: In this scenario, a team of three players (left) tries to score a goal against a single defender and a goalkeeper. The left team starts with the ball and has to dribble past the defender and the goalkeeper to score a goal.

---

[2]`https://github.com/marlbenchmark/on-policy`

Table 15: Hyperparameters in the 2D navigation environment.

| discount | GAE $\lambda$ | PPO epochs | clip parameter | entropy bonus | $\lambda_{\max}$ | actor lr | critic lr | Lagrange lr | batch size |
|---|---|---|---|---|---|---|---|---|---|
| 0.997 | 0.95 | 10 | 0.2 | 0 | 10 | 3e-4 | 1e-3 | 0.5 | 4000 |

Table 16: Common hyperparameters for SIPO, baselines, and ablations.

| discount | GAE $\lambda$ | actor lr | critic lr | clip parameter | entropy bonus | GRF batch size | SMAC batch size |
|---|---|---|---|---|---|---|---|
| 0.99 | 0.95 | 5e-4 | 1e-3 | 0.2 | 0.01 | 9600 | 3200 |

*academy_counterattack_easy*: In this scenario, the left team starts with the ball in the front yard and tries to score a goal against several defenders. All eleven players in the left players can be controlled.

*academy_corner*: In this scenario, the left team tries to score a goal from a corner kick. The right team defends the goal and tries to prevent the left team from scoring. All eleven players in the left players can be controlled.

# D  Implementation Details

## D.1  2D Navigation

We apply PPO with Lagrange multipliers to optimize the policy and hyperparameters are summarized in Table 15. $D(\pi_i, \pi_j)$ is simply taken as the $L_2$ distance of the final state reached by $\pi_i$ and $\pi_j$, i.e., $D(\pi_i, \pi_j) = \|s_H^{\pi_i} - s_H^{\pi_j}\|^2$. The applied algorithm is the same as SIPO (see Appendix G) except that the intrinsic reward is only computed at the last timestep.

## D.2  SIPO

In the $i$-th iteration ($1 \le i \le M$), we learn an actor and a critic with $i$ separate value heads to accurately predict different return terms, including $i - 1$ intrinsic returns for the diversity constraints and the environment reward. We include all practical tricks mentioned in [69] because we find them all critical to algorithm performance. We use separate actor and critic networks, both with hidden size 64 and a GRU layer with hidden size 64. The common hyperparameters for SIPO, baselines, and ablations are listed in Table 16. Other environment-specific parameters, such as PPO epochs and mini-batch size, are all the same as [69]. Besides, Table 14 and Table 17 lists some extra hyperparameters for SIPO.

## D.3  Baselines

We re-implement all baselines with PPO based on the MAPPO [69] project. All algorithms run for the same number of environment frames. Specific hyperparameters for baselines can be found in Appendix D.3.

**SMERL**  SMERL trains a latent-conditioned policy that can robustly adapt to new scenarios. It promotes diversity by maximizing the mutual information between states and the latent variable. We implement SMERL with PPO, where the actor and the critic take as the input the concatenation of observation and a one-hot latent variable. The discriminator is a 2-layer feed-forward network with 64 hidden units. The learning rate of the discriminator is the same as the learning rate of the critic network. The input of the discriminator is the same as the input we use for SIPO-WD. The critic has 2 value heads for an accurate estimation of intrinsic return. Since SMERL trains a

Table 17: SIPO hyperparameters across all environments.

| $\lambda_{\max}$ | Discriminator lr | Lagrangian lr |
|---|---|---|
| 10 | 4.0e-4 | 0.1 |

single latent-conditioned policy, we train SMERL for $M\times$ more environment steps, such that total environment frames are the same. The scaling factor of intrinsic rewards is $0.1$ and the threshold for diversification is $[0.81, 0.45, 0.72]$ ($0.9 \times [0.9, 0.5, 0.8]$) for "3v1", "counterattack", and "corner" respectively.

**DvD** DvD simultaneously trains a population of policies to maximize the determinant of a kernel matrix based on action difference. We concatenate the one-hot actions along a trajectory as the behavioral embedding. The square of the variance factor, i.e., $\sigma^2$ in the RBF kernel, is set to be the length of behavioral embedding. We also use the same Bayesian bandits as proposed in the original paper. Training DvD in "counterattack" and "corner" exceeds the GPU memory and we exclude the results in the main body.

**DIPG** DIPG iteratively maximizes the maximum mean discrepancy (MMD) distance between the state distribution of the current policy and previously discovered policies. For DIPG, we follow the open-source implementation[3]. We set the same variance factor in the RBF kernel as SIPO-RBF and apply the same state as the input of the RBF kernel. We sweep the coefficient of MMD loss among $\{0.1, 0.5, 0.9\}$ and find $0.1$ the most appropriate (larger value will cause training instability). We use the same method to save archived trajectories as SIPO and the input of the RBF kernel is the same as the input we use for SIPO-RBF. To improve training efficiency, we only back-propagate the MMD loss at the first PPO epoch.

**RSPO** RSPO iteratively discovers diverse policies by optimizing extrinsic rewards on novel trajectories while optimizing diversity on other trajectories. The diversity measure is defined as the action-cross entropy along the trajectory. For RSPO, we follow the opensource implementation[4] and use the same hyperparameters on the SMAC *2c_vs_64zg* map in the original paper for GRF experiments.

**TrajDi** TrajDi was originally designed for cooperative multi-agent domains to facilitate zero-shot coordination. It defines a generalized Jensen-Shanon divergence objective between policy action distributions. Then this objective and rewards are simultaneously optimized via population-based training. We tried TrajDi in SMAC and GRF. We sweep the action discount factor among $\{0.1, 0.5, 0.9\}$ and the coefficient of TrajDi loss among $\{0.1, 0.01, 0.001\}$. However, TrajDi fails to converge in the "3v1" scenario and exceeds the GPU memory in the "counterattack" and "corner" scenarios. Therefore, we exclude the performance of TrajDi in the main body.

**Domino** We have meticulously re-implemented Domino within our codebase according to the appendix of Zahavy et al. [71]. We execute the algorithm in the Humanoid locomotion task, employing the robot state (excluding torques) for successor feature computation. Despite our earnest efforts to optimize Domino's performance, our findings reveal its comparable performance to SMERL, illustrated by a minimal diversity score of $0.01$. Therefore, we exclude the performance of Domino in the main body.

**APT** APT maximized the nearest neighbor state-entropy estimation for skill discovery. While we also adopted this metric for diversity evaluation, there is a fundamental distinction in formulation. APT optimizes state entropy within a single policy, whereas our method, SIPO, targets the joint entropy of a population of policies. It is okay for each single policy within the population to have low state entropy. To employ APT's objective of discovering diverse policies, training a population of agents concurrently is required. The algorithm should optimize the estimated entropy over states visited by all policies. Yet, this approach mandates large-scale k-NN computation (k=12) over substantial batches, leading to significant computational inefficiency. Despite our dedicated efforts, we didn't finish a single training trial of APT within 48 hours (in contrast to other PBT baselines, e.g. DvD, which completes training in less than 8 hours).

### D.4 Ablation Study Details

For the three ablation studies: fix-L, CE, and filter, we list the specific hyperparameters here:

---

[3]`https://github.com/dtak/DIPG-public`
[4]`https://github.com/footoredo/rspo-iclr-2022`

- fix-L: we set the Lagrange multiplier to be 0.2;
- CE: the threshold is 3.800 and the intrinsic reward scale factor is $1/1000$ of that in the WD setting;
- filter: all the hyperparameters in the setting are the same as those in the WD setting.

# E    Proofs

## E.1    Proof of theorem 4.1

**Theorem 4.1.** *Assume $D$ is a distance metric. Denote the optimal value of Problem 3 as $T_1$. Let $T_2 = \sum_{i=1}^{M} J(\tilde{\pi}_i)$ where*

$$\tilde{\pi}_i = \arg\max_{\pi_i} \quad J(\pi_i)$$
$$\text{s.t.} \quad D(\pi_i, \tilde{\pi}_j) \geq \delta/2, \quad \forall 1 \leq j < i \tag{3}$$

*for $i = 1, \ldots, M$, then $T_2 \geq T_1$.*

*Proof.* Suppose the optimal solution of Problem 3 is $\pi_1, \pi_2, ..., \pi_M$ satisfying $J(\pi_1) \geq J(\pi_2) \geq ... \geq J(\pi_M)$ and the optimal solution of Problem 6 is $\tilde{\pi}_1, \tilde{\pi}_2, ..., \tilde{\pi}_M$ satisfying $J(\tilde{\pi}_1) \geq J(\tilde{\pi}_2) \geq ... \geq J(\tilde{\pi}_M)$.

Assume the contrary that Thm. 4.1 is not true, which means $\sum_{i=1}^{M} J(\pi_i) = T_1 > T_2 = \sum_{i=1}^{M} J(\tilde{\pi}_i)$. Then we choose the smallest number $N \leq M$ that satisfies

$$\sum_{i=1}^{N} J(\pi_i) > \sum_{i=1}^{N} J(\tilde{\pi}_i).$$

By $T_1 > T_2$ we know that $N$ exists. In addition, because Problem 6 solves unconstrained RL in the first iteration, we know that $\tilde{\pi}_1 = \arg\max_{\pi} J(\pi)$ and then $J(\pi_1) \leq J(\tilde{\pi}_1)$. Therefore, $N \geq 2$.

Suppose $J(\pi_N) \leq J(\tilde{\pi}_N)$. Then we have

$$\sum_{i=1}^{N-1} J(\pi_i) > \sum_{i=1}^{N-1} J(\tilde{\pi}_i).$$

Contradicting the fact that $N$ is the smallest number satisfies that equation.

Hence, we know that $J(\pi_N) > J(\tilde{\pi}_N)$. Then

$$J(\pi_1) \geq J(\pi_2) \geq ... \geq J(\pi_N) > J(\tilde{\pi}_N).$$

Consider the optimization problem of $\tilde{\pi}_N$:

$$\tilde{\pi}_N = \arg\max_{\pi} \quad J(\pi)$$
$$\text{s.t.} \quad D(\pi, \tilde{\pi}_j) \geq \delta/2, \quad \forall 1 \leq j < N.$$

This optimization does not find $\{\pi_1, \ldots, \pi_N\}$ but find $\tilde{\pi}_N$, which means that for each $\pi_i$, $1 \leq i \leq N$, there exists $1 \leq j_i < N$ such that $D(\pi_i, \tilde{\pi}_{j_i}) < \delta/2$. Otherwise, we will get the solution of the above problem as $\pi_i$ instead of $\tilde{\pi}_N$.

By the Pigeonhole Principle, we know that there exist two indexes $i_1 \in [N]$ and $i_2 \in [N]$ ($i_1 \neq i_2$) such that $j_{i_1} = j_{i_2} = \hat{j}$. Then we have

$$D(\pi_{i_1}, \pi_{i_2}) \leq D(\pi_{i_1}, \tilde{\pi}_{\hat{j}}) + D(\pi_{i_2}, \tilde{\pi}_{\hat{j}}) < \delta/2 + \delta/2 = \delta,$$

where the inequality follows by the triangle inequality of the distance function.

It contradict with the fact that $D(\pi_{i_1}, \pi_{i_2}) \geq \delta$ in Problem 3.

Therefore, we prove the theorem $\sum_{i=1}^{M} J(\pi_i) = T_1 \leq T_2 = \sum_{i=1}^{M} J(\tilde{\pi}_i)$. $\qquad\square$

## E.2 Proof of Theorem 5.1

In this section, we consider the $i$-th iteration of SIPO illustrated in Eq. (4). For the sake of simplicity, we use $a \leq \boldsymbol{\lambda} \leq b$ for vector $\boldsymbol{\lambda}$ to denote each component of $\boldsymbol{\lambda}$ satisfies $a \leq \lambda_i \leq b$, where $a, b \in \mathbb{R}$. We use $\pi$ to denote the policy we are optimizing, and $\pi_j$ $(1 \leq j < i)$ to denote a previously obtained policy. We denote the Lagrange function as $L(\pi, \boldsymbol{\lambda}) = -J(\pi) - \sum_{j=1}^{i-1} \lambda_j \left( D(\pi, \pi_j) - \delta \right)$.

To prove Theorem 5.1, we consider the following two optimization problems:

$$(\pi_i, \boldsymbol{\lambda}^\star) = \arg\min_\pi \max_{\boldsymbol{\lambda} \geq 0} L(\pi, \boldsymbol{\lambda}) \tag{11}$$

and

$$(\tilde{\pi}_i, \tilde{\boldsymbol{\lambda}}^\star) = \arg\min_\pi \max_{0 \leq \boldsymbol{\lambda} \leq \Lambda} L(\pi, \boldsymbol{\lambda}), \tag{12}$$

where $\Lambda = \frac{1}{\epsilon_0}$ and $\epsilon_0 > 0$ is sufficiently small.

We make the following assumptions to prove this theorem:

**Assumption E.1.** $0 \leq J(\cdot) \leq 1$.

**Assumption E.2.** $\forall \boldsymbol{\lambda} \geq 0$, $L(\cdot, \boldsymbol{\lambda})$ is $l$-smooth and $\zeta$-Lipschitz.

We may notice that solving the optimization problem (11) is hard because its domain is unbounded. Therefore, we make some approximations and consider the bounded optimization problem (12). First, we prove the following lemma about the value function $J$:

**Lemma E.3.** $J(\pi_i) \leq J(\tilde{\pi}_i)$.

*Proof.* As the domain of $\boldsymbol{\lambda}$ in Eq. 12 is smaller than Eq. (11), we have $L(\pi_i, \boldsymbol{\lambda}) \geq L(\tilde{\pi}_i, \tilde{\boldsymbol{\lambda}})$.

By the fundamental property of Lagrange duality, we know that $L$ achieves its optimal value when $\boldsymbol{\lambda} = 0$ and the optimal value is $-J(\pi_i)$.

By the optimality of $(\tilde{\pi}_i, \tilde{\boldsymbol{\lambda}}^\star)$, we know that

$$-\sum_{j=1}^{i-1} \tilde{\lambda}_j^\star (D(\tilde{\pi}_i, \pi_j) - \delta) \geq 0. \tag{13}$$

Then we have

$$-J(\pi_i) = L(\pi_i, \boldsymbol{\lambda}^\star) \geq \tilde{L}(\tilde{\pi}_i, \tilde{\boldsymbol{\lambda}}^\star) = -J(\tilde{\pi}_i) - \sum_{j=1}^{i-1} \tilde{\lambda}_j^\star (D(\tilde{\pi}_i, \pi_j) - \delta) \geq -J(\tilde{\pi}_i).$$

$\square$

Then we prove the distance between optimal policy $\tilde{\pi}_i$ in problem (12) and optimal policy $\pi_i$ in problem (11) is very small:

**Lemma E.4.** *Under Assumption E.1, $D(\tilde{\pi}_i, \pi_j) \geq \delta - \epsilon_0, \forall 1 \leq j < i$.*

*Proof.* We prove this by contradiction.

Suppose there exists $1 \leq j_0 < i$, $D(\tilde{\pi}_i, \pi_{j_0}) < \delta - \epsilon_0$. Then we choose $\hat{\boldsymbol{\lambda}}$ such that

$$\hat{\lambda}_j = \begin{cases} \Lambda & j = j_0, \\ 0 & 1 \leq j < i, j \neq j_0. \end{cases}$$

By the Assumption E.1, Eq. (13), and $\Lambda = \frac{1}{\epsilon_0}$, we have

$$0 \geq -J(\pi_i) = L(\pi_i, \boldsymbol{\lambda}^\star) \geq L(\tilde{\pi}_i, \tilde{\boldsymbol{\lambda}}^\star) \geq L(\tilde{\pi}_i, \hat{\boldsymbol{\lambda}}) \geq -1 - \Lambda(D(\tilde{\pi}_i, \pi_{j_0}) - \delta) > 0.$$

That is a contradiction. So we have proved that

$$D(\tilde{\pi}_i, \pi_j) \geq \delta - \epsilon_0, \quad \forall 1 \leq j < i.$$

$\square$

From the deduction above, we get the following approximation lemma:

**Lemma E.5.** *Denote the optimal solution of Eq. 11 and Eq. 12 as $(\pi_i, \lambda)$ and $(\tilde{\pi}_i, \tilde{\lambda})$ respectively. Then we have the following approximation about the optimal value and distance:*

$$J(\pi_i) \leq J(\tilde{\pi}_i)$$
$$D(\tilde{\pi}_i, \pi_j) \geq \delta - \epsilon_0, \quad \forall 1 \leq j < i$$

*Proof.* This lemma follows directly by Lemma E.3 and Lemma E.4. □

Therefore, it is reasonable to consider the constrained optimization problem (12) instead of primal problem (11) because we have proved that the optimal value doesn't get smaller and the distance of policy is $\epsilon_0$-approximation of the primal problem. Finally we use the conclusion in the paper [31] to analysis the convergence of problem (12):

**Lemma E.6.** *([31], Theorem 4.8) Under Assumption E.2, solving Eq. (12) via two-timescale GDA with learning rate $\eta_\pi = \Theta(\epsilon^4/l^3\zeta^2\Lambda^2)$ and $\eta_\lambda = \Theta(1/l)$ requires*

$$\mathcal{O}\left(\frac{l^3\zeta^2\Lambda^2 C_1}{\epsilon^6} + \frac{l^3\Lambda^2 C_2}{\epsilon^4}\right)$$

*iterations to converge to an $\epsilon$-stationary point $\pi_i^\star$, where $C_1$ and $C_2$ are the constants that depend on the distance between the initial point and the optimal point.*

**Theorem 5.1.** *Under assumptions E.1 and E.2 and learning rate with learning rate $\eta_\pi = \Theta(\epsilon^4/l^3\zeta^2\Lambda^2)$ and $\eta_\lambda = \Theta(1/l)$, SIPO converges to an $\epsilon$-stationary point with convergence rate $\mathcal{O}\left(\frac{l^3\zeta^2\Lambda^2 C_1}{\epsilon^6} + \frac{l^3\Lambda^2 C_2}{\epsilon^4}\right)$.*

*Proof.* We consider the following constraint nonconvex-concave optimization:

$$\min_\pi \max_{0 \leq \boldsymbol{\lambda} \leq \Lambda} L(\pi, \boldsymbol{\lambda}). \tag{14}$$

Following Lemma E.6, we know that the Two-Timescale GDA algorithm converges to an $\epsilon$-stationary point $\pi_i^*$.

From the above deduction, the Two-Timescale GDA algorithm requires $\mathcal{O}\left(\frac{l^3\zeta^2\Lambda^2 C_1}{\epsilon^6} + \frac{l^3\Lambda^2 C_2}{\epsilon^4}\right)$ iterations with learning rate $\eta_\pi = \Theta(\epsilon^4/l^3\zeta^2\Lambda^2)$ and $\eta_\lambda = \Theta(1/l)$ to converge to an $\epsilon$-stationary point with convergence rate.

□

# F  Discussion

## F.1  The Failure Case of State-Distance-Based Diversity Measures

A failure case of state-distance-based diversity measures may be when the state space includes many *irrelevant features*. These features cannot reflect behavioral differences. If we run SIPO in such an environment, the learned strategies may be only diverse w.r.t these features and have little visual distinction. Like the famous noisy TV problem [5], the issue of irrelevant features is intrinsically challenging for general RL applications, which cannot be resolved by using action-based or state-occupancy-based diversity measures either.

Thanks to the advantages we discussed in the paper, we generally find that state-distance-based measures can be preferred in challenging RL problems. Meanwhile, since the state dimension can be much higher than actions, it is possible that RL optimization over states may be accordingly more difficult than actions. In practice, we can design a feature selector for those most relevant features for visual diversity and run diversity learning over the filtered features. In SMAC and GRF, we utilize the agent features (excluding enemies) as the input of diversity constraint without further modifications, as discussed in Appendix D. We remark that even after filtering, the agent features remain high-dimensional while our algorithm still works well. Note that using a feature selector is a common practice in many existing domains, such as novelty search [12], exploration [33], and curriculum learning [7]. There are also works studying how to extract useful low-dimensional features from observations [68, 18], which are orthogonal to our focus.

## F.2  The Distance Metric

In Sec. 5, we adopt the two most popular implementations in the machine learning literature, i.e., RBF kernel and Wasserstein distance, while it is totally fine to adopt alternative implementations. For example, we can learn state representations (e.g. auto-encoder, Laplacian, or successor feature) and utilize pair-wise distance or norms as a diversity measure. Similar topics have been extensively discussed in the exploration literature [68, 41]. We leave them as our future directions.

## G  Pseudocode of SIPO

The pseudocode of SIPO is shown in Algorithm 1.

---

**Algorithm 1** SIPO (red for SIPO-RBF and blue for SIPO-WD)

---

**Input:** Number of Iterations $M$, Number of Training Steps within Each Iteration $T$.
**Hyperparameter:** Learning Rate $\eta_\pi$, Diversity Threshold $\delta$, Intrinsic Scale Factor $\alpha$, Lagrange Multiplier Upperbound $\lambda_{\max}$, Lagrange Learning rate $\eta_\lambda$, Wasserstein Critic Learning Rate $\eta_W$, RBF Kernel Variance $\sigma$.

1: Archived trajectories $X \leftarrow \emptyset$      // to store states visited by previous policies
2: **for** iteration $i = 1, \ldots, M$ **do**
3:      Initialize policy $\pi_{\theta_i}$      // initialization
4:      Initialize Wasserstein critic $f_{\phi_i}$
5:      **for** archive index $j = 1, \ldots, i - 1$ **do**
6:          Lagrange multiplier $\lambda_j \leftarrow 0$
7:      **end for**
8:      **for** Training step $t = 1, \ldots, T$ **do**
9:          Collect trajectory $\tau = \{(s_h, \boldsymbol{a}_h, r(s_h, \boldsymbol{a}_h))\}_{h=1}^H$
10:         **for** archive index $j = 1, \ldots, i - 1$ **do**
11:            $R_{\text{int}}^j \leftarrow 0$
12:         **end for**
13:         **for** timestep $h = 1, \ldots, H$ **do**
14:            $r_{\text{int},h} \leftarrow 0$      // compute intrinsic reward
15:            **for** archive trajectory $\chi_j \in X$ **do**
16:              $r_{\text{int},h}^j \leftarrow -\frac{1}{H|\chi_j|} \sum_{s' \in \chi_j} \exp\left(-\frac{\|s_h - s'\|^2}{2\sigma^2}\right)$
17:              $r_{\text{int},h}^j \leftarrow \frac{1}{H}\left[f_{\phi_j}(s_h) - \frac{1}{|\chi_j|}\sum_{s' \in \chi_j} f_{\phi_j}(s')\right]$
18:              $r_{\text{int},h} \leftarrow r_{\text{int},h} + \lambda_j \cdot r_{\text{int},h}^j$
19:              $R_{\text{int}}^j \leftarrow R_{\text{int},h}^j + r_{\text{int},h}^j$
20:            **end for**
21:            $r_h \leftarrow r(s_h, \boldsymbol{a}_h) + \alpha \cdot r_{\text{int},h}$
22:         **end for**
23:         **for** archive index $j = 1, \ldots, i - 1$ **do**
24:            $\lambda_j \leftarrow \text{clip}\left(\lambda_j + \eta_\lambda\left(-R_{\text{int}}^j + \delta\right), 0, \lambda_{\max}\right)$      // gradient ascent on $\lambda_j$
25:            $\phi_j \leftarrow \phi_j + \eta_W \frac{1}{H}\sum_{h=1}^H \nabla_{\phi_j}\left(f_{\phi_j}(s_h) - \frac{1}{|\chi_j|}\sum_{s' \in \chi_j} f_{\phi_j}(s')\right)$
26:            $\phi_j \leftarrow \text{clip}(\phi_j, -0.01, 0.01)$
27:         **end for**
28:         Update $\pi_{\theta_i}$ with $\{(s_h, \boldsymbol{a}_h, r_h)\}$ by PPO algorithm      // policy gradient on $\theta_i$
29:      **end for**
30:      Collect many trajectories $\chi_i$      // collect trajectories to approximate $d_{\pi_{\theta_i}}$
31:      $X \leftarrow X \cup \{\chi_i\}$      // for the use of following iterations
32: **end for**

---

