# OpenReview forum: "Iteratively Learn Diverse Strategies with State Distance Information"
_NeurIPS.cc/2023/Conference — NeurIPS 2023 poster_

### Official Review · Reviewer_hxqg · 2023-06-28

**Soundness:** 3 good
**Presentation:** 2 fair
**Contribution:** 2 fair
**Rating:** 6
**Confidence:** 3

**Summary:**

Reinforcement Learning (RL) algorithms commonly learn a distinct policy that is responsible for a distinct behavior. Learning different, diverse behaviors generally is a difficult task in RL. This paper proposes a new algorithm State-based Intrinsic-reward Policy Optimization (SIPO) that can learn diverse, human interpretable behaviors.
The paper first discusses some of the current diversity measures, and based on the insight, it proposes the new ITR-based method, even though PBT is the choice when it comes to learning diverse behaviors. The method is efficient and can learn diverse behaviors. The effectiveness is demonstrated in simulated environments.

**Strengths:**

- well written. Motivation and contributions made clear
- the paper covers a good amount of related work, even though there are a few missing, in my opinion (see Weaknesses).
- Nice evaluations w.r.t. diversity

**Weaknesses:**

- The big field of unsupervised reinforcement learning also covers a lot of similarity/dissimilarity measures for formulating an intrinsic reward. Even though some of the works are mentioned (e.g. DIAYN), there are several other works that should be mentioned as they are using different objectives and hence differently measure the similarity/dissimilarity:
	- https://arxiv.org/pdf/1906.05274.pdf
	- http://proceedings.mlr.press/v139/yarats21a/yarats21a.pdf

During a Google search, I discovered another work that also seems to have an interesting objective to learn different solutions:
	- https://proceedings.mlr.press/v164/celik22a/celik22a.pdf

- weak motivation for the necessity of the state distance (see Questions)

**Questions:**

- The motivation behind Figure 2 is unclear to me even when reading the text. I think this figure definitely needs more description and explanation. There are also no numbers indicating what the measured distances are.

- Different works have approximated the state entropy for discovering new skills. E.g. the APT [1]. It would be interesting to see how the used diversity measures compare against it. Can the authors assess, how different the solutions would be when using the approximated state entropy compared to the proposed metrics?

- The humanoid locomotion task in Section 6.1 is used to compare the diversity of learned policies. How do the methods compare w.r.t. performance?

[1]: https://openreview.net/forum?id=fIn4wLS2XzU

**Limitations:**

The paper explicitly addresses limitations but doesn't state anything about potential negative societal impact.

---

> ### Author Rebuttal · Authors · 2023-08-10
>
> +  there are several other works that should be mentioned as they are using different objectives and hence differently measure the similarity/dissimilarity
>
> We appreciate the reviewer's insightful suggestion and ensure that we will incorporate these relevant works into our paper. The reviewer’s additional reference[1] operates in a parallel setting where a mixture of policies is learned within task-aware contextual MDPs, with each policy addressing a subspace of tasks. In contrast, our focus is on the discovery of distinct individual policies to tackle the same task.
>
> [1] Celik, O., Zhou, D., Li, G., Becker, P., & Neumann, G. (2022, January). Specializing versatile skill libraries using local mixture of experts. In Conference on Robot Learning (pp. 1423-1433). PMLR.
>
> + The motivation behind Figure 2 is unclear to me even when reading the text. I think this figure definitely needs more description and explanation. There are also no numbers indicating what the measured distances are.
>
> As an example, in the soccer attack, the goalkeeper could misleadingly amplify the action-based diversity score by outputting random actions (e.g. sliding in the backyard). This example underscores a notable issue. If action-based measures are leveraged for optimizing diversity, the resultant policies can produce visually similar behavior.
>
> + It would be interesting to see how the used diversity measures compare against APT.
>
> We appreciate the suggestion. However, there is a fundamental distinction in formulation. APT optimizes state entropy within a *single policy*, whereas our method, SIPO, targets the joint entropy of *a population of policies*. It is okay for each single policy within the population to have low state entropy.
>
> To employ APT's objective, training a population of agents concurrently is required. The algorithm should optimize the estimated entropy over states visited by all policies. Yet, this approach mandates large-scale k-NN computation (k=12) over substantial batches, leading to significant computational inefficiency. Despite our dedicated efforts, we didn’t finish a single training trial of APT within 48 hours (in contrast to other PBT baselines, e.g. DvD, which completes training in less than 8 hours). We primarily employ the k-NN state entropy metric to quantify diversity in the SMAC domain, as we show in the paper.
>
> + The humanoid locomotion task in Section 6.1 is used to compare the diversity of learned policies. How do the methods compare w.r.t. performance?
>
> We present the diversity score and corresponding average rewards achieved by all algorithms below. These numerical values are averaged across the complete population for a clear comparison:
>
> | 		|SIPO-RBF | SIPO-RBF | DIPG | RSPO | SMERL | DvD | PPO |
> | --- | --- | --- | --- | --- | --- | --- | --- |
> | humanoid reward |3508 	| 3763 		| 5191 | 1455 |4253 | 4498 | 5299 |
> | humanoid diversity | 0.53 | 0.71 		| 0.12 | 	0.53 | 0.01 | 0.40 | -	|
>
> The tabulated data above highlights the varying trade-offs between task performance and diversity exhibited by different algorithms. It is noteworthy that SIPO, in particular, displays an adeptness at training a notably more diverse population while upholding a reasonably moderate level of task performance.

---

> > ### Comment · Reviewer_hxqg · 2023-08-19
> > **Answer to the authors**
> >
> > I am sorry for my late reply. I am thankful for the author's detailed explanations. I don't have further questions and would like to wait for the discussion period for further action. However, the author's response helps me to recommend a 'weak accept'.

---

### Official Review · Reviewer_3iHV · 2023-06-30

**Soundness:** 2 fair
**Presentation:** 3 good
**Contribution:** 2 fair
**Rating:** 4
**Confidence:** 4

**Summary:**

This paper addresses the problem of learning diverse policies for a given RL task. First, it studies pros and cons of various formulations in two different dimensions: (i) How to measure diversity, for which it considers measures based on action or state distribution and state distances, (ii) how to compute the diverse policies, either with a joint method or an iterative one. Then, it combines the findings into a method, called SIPO, which implements iterative learning through gradient descent ascent with state distance incentives. Finally, it empirically evaluates SIPO on a variety of domains, including locomotion tasks, Star-Craft, and Google Research Football.

**Strengths:**

- Relevance of the problem. Several recent works have addressed the problem of learning diverse policies in RL, in both supervised and unsupervised settings;
- Methodology. Interesting and reasonably efficient methodology that is based on a new state-metric perspective on the problem of learning diverse policies;
- Experimental results. Promising empirical results on challenging domains.

**Weaknesses:**

- Motivation. The introduced objective lacks a strong theoretical ground that is mostly present in previous works with alternative diversity objectives;
- Related works. The comparison between SIPO and some relevant related works, especially Zahavy et al., 2022, is not really upfront;
- Experiments robustness and significance. The experiments consider arbitrary performance metric and average results over 5 or 3 seeds, which is hardly enough to get statistical significance.

**Questions:**

I have one crucial concern about this paper, which is otherwise sound and interesting. The superiority of diversity measures based on state distances is only informally motivated, through examples rather than theoretical justification. Instead, the diversity measures based on action distribution or state occupancy have been both linked to important theoretical properties, such as reward robustness for the former (e.g., Husain et al., Regularized policies are reward robust, 2021) or bounded sub-optimality for the latter (e.g., Kakade and Langford, Approximately optimal approximate reinforcement learning, 2022). Diversity based on state distances currently lacks similar theoretical ground, which also leaves an existential question on the motivation of the paper? Why we want to learn diverse strategies in the first place? Is it to be robust to reward perturbations? Is to learn reusable skills? Is it something else?

The concern reported above somewhat propagates to the experimental evaluation. If the ultimate purpose of the training is not clear, how can we evaluate one method against another? Indeed, the experiments are based on arbitrary diversity scores. SIPO may be better than baselines on those, but it is hard to reach some conclusion beyond qualitative justifications. Perhaps a good option would be to evaluate SIPO also on more consequential benchmarks, such as testing the reward robustness of the trained set of policies, or showing that the latter is a good starting point for fine-tuning to a different task.

For this main reason, I am currently providing a borderline evaluation, while I am open to change my score if the authors can provide more formal justification for their objective function in the rebuttal.

I provide below additional comments on the paper.

**Objective function**

One key aspect that Section 4.1 seems to overlook is that all of the provided motivations (especially in lines 158-169) make sense only when the environment's states lie on a metric space. In several domains, such as tabular MDPs with symbolic state representations or vision-based tasks, defining a proper metric over the states might be even more challenging than learning diverse policies alone. I think this limitation should be explicitly reported everywhere in the paper, including the introduction and abstract.

The motivation reported in lines 170-173 appears to be somewhat weak too, at least in domains where external rewards are present, and the concept of idle actions may be incorporated in the rewards.

Can the authors clarify the role of the cost function $g$ into the diversity score reported in Eq. 5?

**Experiments**

Why are the authors reporting experiments in multi-agent domains, even if they present a method for learning diverse policies in a single-agent setting? This choice looks somewhat odd, and it shall be motivated.

I am worried about the statistical significance of the reported performance results, since most of them are obtained as average over 5 or 3 seeds. While I can understand that running experiments in such complex domains is costly, the standard to claim experimental robustness has increased lately. Moreover, the results report average and standard deviation. On the one hand, confidence intervals would be more meaningful than standard deviation, on the other hand only it is not enough to have best average performance to say that one method outperforms another when the intervals are overlapping.

**Related Works**

This work looks in a way orthogonal to Zahavy et al. (2022), in which the authors present a method to learn diverse near-optimal policies by maximizing diversity with an hard reward constraint. While their concept of diversity is different, it would be interesting to discuss pros and cons of the two alternative solutions in depth, and perhaps compare SIPO with Domino in the experimental campaign.

Another work that is somewhat relevant to the topic is (Mutti et al., Reward-free policy space compression for reinforcement learning). Their aim is to learn a diverse set of policies from which any reward function can be approximately optimized, and they also present a gradient descent ascent procedure for this purpose. The fact that their work does not account for rewards is an important difference, but the authors can perhaps consider discussing this work in the paper as well.

One key benefit that I see in the SIPO solution w.r.t. prior works, is that SIPO shall enjoy favorable computational complexity, whereas both Zahavy et al. and Mutti et al. need to solve a non-convex non-concave optimization problem.

**Minor**

- The preliminaries are framing the problem as POMDP. I do not understand why partial observability is introduced, and then never used in the paper at all.
- Figure 4, IL is instead ITR?
- Theorem 4 says that ITR can achieve the same rewards as PBT, but only with an inferior diversity. This trade-off between computation and diversity could be better highlighted.
- One of the evaluation metric is the entropy estimator via k-NN, for which (Liu & Abbeel, 2021) is mentioned. Note that those entropy estimators have been presented before (e.g., Singh et al., Nearest neighbor estimates of entropy, 2001). Moreover, if the goal is to maximize the state entropy, why not using APT instead of SIPO?

**Limitations:**

The limitations are reported in the final sentences of the paper. Perhaps, some limiting aspects could be discussed in additional length.

---

> ### Author Rebuttal · Authors · 2023-08-10
>
> + motivation
>
> Our primary aim is to achieve diversity itself. Agents with similar reward functions can manifest significantly diverse behaviors (e.g., high-reward unexpected behavior[1]). This property is different from standard DL where different local optima (almost) suggest the same results (i.e., proxy to the global optimum[2]).
>
> This prompts the importance of studying *the learned behavior* and finding *all unique solutions for the same task* in RL beyond only the rewards. We believe it is a **fundamental research problem in RL**, which, however, very few researchers have been fully aware of. A diverse collection of policies can further enhance human-AI collaboration or help design robust robots, depending on the applicaion (line27-line30).
>
> Despite our best efforts to engage in theoretical analysis, we note that a consensus of the best algorithmic formulation of "distinct solutions" in RL remains elusive.
>
> [1] https://openai.com/blog/ 409 faulty-reward-functions/.
>
> [2] Ma, T. (2020). Why Do Local Methods Solve Nonconvex Problems?.
>
> + clarification of the evaluation metric
>
> We endeavored to select the best available task-specific metric to authentically reflect behavioral diversity. For humanoid, we adopt a metric embraced by the Quality-Diversity community. However, for SMAC and GRF, existing metrics can fail (e.g. Table 8 in [1] and our Appendix B.2.3). Therefore, it is necessary to engineer task-specific metrics relying on both visualization-based insights and metrics in previous works.
>
> We enthusiastically encourage the reviewer to review all replay files utilized for tallying policy counts on our project website: https://mega.nz/folder/xP0D2CRa#YRL-PVjjsyZhGZ2QUZqH2g.
>
> [1] https://arxiv.org/abs/2204.02246
>
> + tabular MDPs/vision-based tasks
>
> We explicitly assume the access to object-centric information and features in our paper. We emphasize that this assumption typically holds in MARL benchmarks and can be further addressed by incorporating feature learning algorithms, e.g. [1] and [2]. We discussed this in Appendix F.2 and are committed to reinforcing this.
>
> We note that SIPO-WD has already addressed this assumption through the incorporation of a learnable Wasserstein discriminator. The discriminator can effectively process images or one-hot state embeddings as inputs, as detailed in Appendix B.3.
>
> [1] https://arxiv.org/abs/1810.04586
>
> [2] https://arxiv.org/abs/2102.11271
>
> + The motivation in lines 170-173 appears weak
>
> As an example, in the soccer attack, the goalkeeper could misleadingly amplify the action-based diversity score by outputting random actions (e.g. sliding in the backyard). This example underscores a notable issue. If action-based measures are leveraged for optimizing diversity, the resultant policies can produce visually similar behavior.
>
> While it can be possible to exclude idle actions by modifying task rewards, it requires a large number of hacks and engineering efforts. The issue of idle actions exists even in popular MARL benchmarks like GRF. We propose a systematic approach to bypass idle actions by directly considering state distances.
>
> + the role of the cost function g?
>
> The cost function g is a notation providing a generalized and unified definition. It also contributes to training stability by scaling the raw distance. g in Eq.5 can be realized by either RBF kernel or Wasserstein distance.
>
> + multi-agent experiments and POMDP
>
> We wish to validate that SIPO is general enough to be applied to many difficult and complicated scenarios, such as testbeds like GRF and SMAC (SMAC has partial observability). These environments encompass a notably more diverse range of potential winning strategies, and existing methods can fail. Therefore, they offer an apt platform for assessing SIPO's capacity.
>
> (Dec-)POMDP is a standard formulation in many MARL works. We follow this common definition since it provides the fewest algorithmic assumptions.
>
> + Comparison with Domino
>
> While we have discussed in Sec 2, we are delighted to present an in-depth discussion below.
>
> **Formulation** Domino pursues hard constraints on rewards while optimizing diversity, potentially hindering policies with disparate reward scales. Therefore, Domino tends to discover similar policies but all with the same reward. SIPO, on the other hand, allows much diverse locally optimal policies.
>
> **Diversity Measure** Domino employs the distance between successor features, while SIPO directly utilizes the distance between raw states. Domino is complementary to our worker because SIPO can also incorporate successor features as the state representation.
>
> We have meticulously re-implemented Domino within our codebase according to Domino’s appendix. We execute the algorithm in the Humanoid locomotion task, employing the robot state (excluding torques) for successor feature computation. Despite our earnest efforts to optimize Domino's performance, our findings reveal its comparable performance to SMERL, illustrated by a minimal diversity score of 0.01.
>
> + APT metric and its usage in experiments
>
> We acknowledge the missing reference and will duly append it to our paper.
>
> Regarding the utilization of APT, we appreciate the suggestion. However, there is a fundamental distinction in formulation. APT optimizes state entropy within a *single policy*, whereas our method, SIPO, targets the joint entropy of *a population of policies*. It is okay for each single policy within the population to have low state entropy.
>
> To employ APT's objective, training a population of agents concurrently is required. The algorithm should optimize the estimated entropy over states visited by all policies. Yet, this approach mandates large-scale k-NN computation (k=12) over substantial batches, leading to significant computational inefficiency. Despite our dedicated efforts, we didn’t finish a single training trial of APT within 48 hours (in contrast to other PBT baselines, e.g. DvD, which completes training in less than 8 hours).

---

> > ### Comment · Reviewer_3iHV · 2023-08-18
> > **After response**
> >
> > I want to thank the authors for their detailed replies.
> >
> > Unfortunately, I am still doubtful on both the strength of the motivation and the choice of evaluation metrics. I concede that those are matter of opinion rather than formal/technical weaknesses. I see little chance of solving them through back-and-forth discussion. Instead, I will hear other opinions in the private discussion before making a final evaluation.
> >
> > As a minor note, I think that proposing SIPO as a way to "find all unique solutions for the same task in RL" is somehow overstated. This looks closer to the objective of Domino, in which one maximizes the diversity of a set of nearly optimal policies. Instead, SIPO does not control the sub-optimality, as it has an hard constraint on diversity. This means that some of the policies provided by SIPO are not solutions to the task, even in approximate sense.

---

> > > ### Author Response · Authors · 2023-08-19
> > > **Additional Author Response**
> > >
> > > We appreciate the reviewer's active engagement. We believe that your insights and feedback are immensely valuable to us. Thanks a lot.
> > >
> > > We also have a brief reply to the minor comment regarding the difference between Domino and SIPO. We focus more on the interesting emergent behaviors than the reward score. We want to emphasize that **in the multi-agent setting**, policies with lower scores **can still be** solutions to the problem. It can be proved that each local optima in cooperative multi-agent games is a global nash equilibrium[2], which is a meaningful solution to the problem. Domino has its own flaw that it can easily ignore a significant collection of solutions (e.g. in multi-agent trust dilemmas[1]).
> > >
> > > [1] Peysakhovich, A., & Lerer, A. (2017). Prosocial learning agents solve generalized stag hunts better than selfish ones. AAMAS 2018.
> > >
> > > [2] Emmons, S., Oesterheld, C., Critch, A., Conitzer, V., & Russell, S. (2022, June). For learning in symmetric teams, local optima are global nash equilibria. In International Conference on Machine Learning (pp. 5924-5943). PMLR.

---

### Official Review · Reviewer_hytR · 2023-07-05

**Soundness:** 3 good
**Presentation:** 3 good
**Contribution:** 2 fair
**Rating:** 6
**Confidence:** 4

**Summary:**

The paper discusses the challenge of optimizing rewards while discovering diverse strategies in complex reinforcement learning problems. This paper examines two design choices for tackling this challenge: diversity measure and computation framework. By incorporating state-space distance information into the diversity measure, the behavioral differences between policies are accurately captured. Besides, two common computation frameworks: population-based training (PBT) and iterative learning (ITR) are compared, and it shows that ITR can achieve comparable diversity scores with higher computation efficiency. Based on above analysis, a novel diversity-driven RL algorithm named State-based Intrinsic-reward Policy Optimization (SIPO) is proposed. The authors evaluate SIPO across three environments and demonstrate that it consistently produces diverse policies that cannot be discovered by existing baselines.

**Strengths:**

This paper introduces a state-based population diversity measurement and transfers it into a shaping reward in practice for computational conveniency. By comparison with PBT and ITR, it is concluded that ITR is able to achieve higher performance. Additionally, experiment environments including single-agent reinforcement learning and multi-agent reinforcement learning all shows the effectiveness of the proposed method. In a way, this paper is well-written and presents some contributions to the field of reinforcement learning. The authors provide clear explanations of their methodology and experiment results, making the paper easy to track.

**Weaknesses:**

In this paper, the proposed state-based diversity measurement is one of key innovations. By combining with ITR framework, it achieves satisfactory performance. However, there still exist some weaknesses remained to be polished up.

(1) The comprehensiveness of literature reviews remains to be improved.

(2) The motivating example in Sec4.1 shows the limitation of previous action-based diversity measurement. However, the reason why state-distance based measurement can overcome the problem is not explained adequately.

(3) The experiment part seems to be insufficient. Some SOTA diversity enhancement methods in RL and MARL are not analyzed or summarized. Besides, the assessment criteria of diversity seems unfair.

(4) It claims that the heatmaps of agent positions in SMAC show obvious strategic differences. However, all four policies seem to explore similar areas in the map.

(5) While the main body of the paper is well-written, there is space for improvement. I defer some of my issues in the appendix to "Questions".


**Questions:**

Q1: Would you improve the literature reviews from the aspect of adding some well-known or SOTA related works (e.g. [1] [2] [3] [4] [5])?

Q2: Why can state-distance based measurement overcome the shortcomings of action-based measurements? It is suggested to make more discussion theoretically.

Q3: Can SIPO be extended to algorithms in two-player zero-sum game scenario, like self-play, PSRO?

Q4: What will be like if combining state-based diversity measurement and action-based diversity measurement? It is suggested to add an study in experiment part.

Q5: Since the number of population size is fixed in PBT or ITR, the determination of it is still an open question as we known. Would you add a theoretical or empirical analysis on it?

Q6: Comparison criteria of diversity seems unfair to baselines in Humanoid Locomotion and SMAC, since all baselines’ optimization objective are action-based. An intuitive evaluation criteria is the quantifiable behavioral difference between agents. Could you make more analysis on it?

Q7: As can be seen, the heatmaps in Fig7 seems to show that the positions explored by the four agents are not significantly different. Would you add additional explanations on it?

Q8: Minor: abbreviation in Fig4 (i.e., “IL”) is not been explained.

[1] Liu, Z., Yu, C., Yang, Y., Wu, Z., & Li, Y. (2022). A Unified Diversity Measure for Multiagent Reinforcement Learning. Advances in Neural Information Processing Systems, 35, 10339-10352.

[2] Balduzzi, D., Garnelo, M., Bachrach, Y., Czarnecki, W., Perolat, J., Jaderberg, M., & Graepel, T. (2019, May). Open-ended learning in symmetric zero-sum games. In International Conference on Machine Learning (pp. 434-443). PMLR.

[3] Hu, S., Xie, C., Liang, X., & Chang, X. (2022, June). Policy diagnosis via measuring role diversity in cooperative multi-agent rl. In International Conference on Machine Learning (pp. 9041-9071). PMLR.

[4] Masood, M. A., & Doshi-Velez, F. (2019). Diversity-inducing policy gradient: Using maximum mean discrepancy to find a set of diverse policies. arXiv preprint arXiv:1906.00088.

[5] Li, C., Wang, T., Wu, C., Zhao, Q., Yang, J., & Zhang, C. (2021). Celebrating diversity in shared multi-agent reinforcement learning. Advances in Neural Information Processing Systems, 34, 3991-4002.


**Limitations:**

The authors have concluded the limitation of the methodology from the aspect of state representation and acceleration problem.

---

> ### Author Rebuttal · Authors · 2023-08-10
>
> + Some SOTA diversity enhancement methods in RL and MARL are not analyzed or summarized.
>
> We sincerely thank the reviewer for providing additional relevant literature, but we believe that our paper does well in contrasting against previous methods for developing diverse policies in RL.
>
> [1] and [2] study diversity in competitive multi-agent games for minimizing exploitability. This objective does not apply to our settings, i.e., single-agent MDPs and cooperative multi-agent games.
>
> [3] and [5] explore cooperative multi-agent games, gauging diversity **among agents within each game**. Their final objectives are optimizing rewards. In contrast, our approach focuses on diversity across **a collection of distinct joint policies**. The final reward is the secondary objective in our setting. [5] has been discussed in line 89 of our paper. We commit to incorporating [3] in our forthcoming revision.
>
> We acknowledge the significance of [4] (i.e., DIPG) in relation to our paper. This work is extensively discussed in our related work section and comprehensively juxtaposed in our experimental analyses (cited as [39] in our paper).
>
> + Why can state-distance based measurement overcome the shortcomings? Theoretical discussion?
>
> Action-based measures may fail when visually similar states are reached through very different action sequences. We provide particular **counterexamples** of action-based measures in Sec 4.1, suggesting fundamental flaws of existing methods.
>
> State-distance based measures circumvent this issue by explicitly comparing states, which directly addresses the counter example. Based on this finding, we further empirically validate our proposed methods on many challenging domains with empirical results in Sec 6. How to best quantify diversity remains an open question. A widely accepted theoretical framework has not yet converged in the community. For this reason, we choose to analyze the algorithmic issues using specific examples and an analysis on algorithm convergence.
>
>
> + Can SIPO be extended to algorithms in self-play, PSRO?
>
> SIPO's design inherently enables the generation of diverse policies for conquering static opponents, as illustrated by SMAC and GRF experiments. We acknowledge that the integration of SIPO (low-level policy solver for best responses) with PSRO (high-level policy mixture) holds great potential. We regard this as a promising avenue for future exploration.
>
> + What will be like if combining state-based diversity measurement and action-based diversity measurement?
>
> The reviewer’s suggestion is greatly appreciated. In response, we have carried out additional experiments within the GRF domain to explore this approach. We introduce action information by directly concatenating the global state, used for diversity calculation, with the one-hot encoded actions of all agents. The following table presents the outcomes, indicating the number of policies obtained:
>
> |	|		3v1 | CA | corner |
> | --- | --- | ---| ---|
> | SIPO-RBF | 3.0 (0.8) | 3.3 (0.5) | 2.7 (0.5) |
> | SIPO-RBF + action | 3.0 (0.0) | 2.3 (0.5) | 1.0 (0.0) |
>
> For scenarios with a limited number of agents, the action-augmented variant demonstrates comparable performance. However, when the agent count increases (as evident in the 11-agent cases of CA and corner), the incorporation of actions can introduce misleading diversity, detracting from the authenticity of the outcomes. We will append these results in our final version.
>
> + the determination of the population size
>
> A larger population usually leads to a better performance on the desired application[1]. SIPO/ITR enables open-ended training for diversity discovery. With unlimited time and computation, training can proceed until no additional distinct strategies can be discovered. In our paper, we run with the maximal population size given available resources.
>
> [1] Tang, Z. et al. (2021). Discovering diverse multi-agent strategic behavior via reward randomization. ICLR.
>
> + Comparison criteria of diversity seems unfair to baselines in Humanoid Locomotion and SMAC
>
> The adopted criterion (i.e., torque distance, k-NN state entropy) is a commonly used evaluation metric[2,4]. Prior works (i.e., our baselines) develop diversity measures for their own purposes, such as encouraging exploration and deriving robust policies, which may fail to learn meaningful diverse policies. Our measures are designed to directly tackle this diversity problem based on the analysis of counter-examples and empirical evidence. Adopting a state-based criterion does not mean that SIPO will benefit from it and lead to an unfair evaluation since no methods explicitly optimize this evaluation criterion.
>
> Prior works[3] have also attempted to use an action-based metric, i.e., DvD score. However, [3] reported that this criteria can easily achieve the maximum value of 1.0 and does not show distinction among different algorithms.
>
> [2] Shuang Wu, Jian Yao, Haobo Fu, Ye Tian, Chao Qian, Yaodong Yang, QIANG FU, and Yang Wei. Quality-similar diversity via population based reinforcement learning. In The Eleventh International Conference on Learning Representations, 2023.
>
> [3] Zhou, Z., Fu, W., Zhang, B., & Wu, Y. (2022). Continuously discovering novel strategies via reward-switching policy optimization. arXiv preprint arXiv:2204.02246.
>
> [4] Singh et al., Nearest neighbor estimates of entropy, 2001
>
> + As can be seen, the heatmaps in Fig7 seems to show that the positions explored by the four agents are not significantly different. Would you add additional explanations on it?
>
> To facilitate a comprehensive comparison, we have included policies generated by DIPG for the reviewer's consideration in the attached pdf. Compared with DIPG, we contend that the heatmap in our paper exhibits greater behavioral diversity. Moreover, we strongly encourage the reviewer to peruse the GIF demonstrations on our [project website](https://sites.google.com/view/diversity-sipo) for an enhanced qualitative assessment.

---

> > ### Comment · Reviewer_hytR · 2023-08-17
> >
> > First of all, thank you to the author for response to the questions which I raised. Some concerns have been answered adequately. According to the author's response, I agree to increase the score of this article by 2 point to "weak accept".

---

### Official Review · Reviewer_tun3 · 2023-07-06

**Soundness:** 4 excellent
**Presentation:** 4 excellent
**Contribution:** 3 good
**Rating:** 8
**Confidence:** 3

**Summary:**

This work proposes a solution to the problem of finding diverse policies for complex (multi-agent) reinforcement learning (RL) environments. The paper is presented as a joint study on diversity metrics and learning frameworks. For the former, the authors show the limitations of common diversity metrics like action-distribution diversity and state-occupancy based metrics and argue for metrics that incorporate state-distances for training mutually-distinct policies instead. For the latter, population-based (PBT) and iterative training (ITR) approaches are considered, where PBT is presented as a more formally suitable framework, that in reality is limited by its pairwise diversity constraints. ITR as a reasonable relaxation of PBT, on the other hand, is found to be effective in combination with the author’s proposed (’meaningful’) state-distance diversity. The combination if ITR and the proposed metric is incorporated into the State-Based Intrinsic-Reward Policy Optimization (SIPO) algorithm and evaluated on the single-agent human locomotion domain, as well as the multi-agent SMAC and GRF environments. SIPO is shown to outperform related baselines and is presented to be more capable of producing visually distinct / humanly intuitive policies compared to its counterparts.

**Strengths:**

- The paper is well researched in related work. While the evaluated frameworks PBT and ITR or the concept of state-distance in itself are not novel (but well cited and explained), the combination of ITR with the proposed state-dissimilarity diversity -- realized either as RBF-Kernel or with Wasserstein-Distance — are novel as far as I can tell.
- Both the study on the limitations and the evaluation are presented in-depth and read quite intuitive. I am not too familiar with formal proofs of convergence criteria but the technical aspects in the paper itself do appear to be sound. The evaluation is reasonably complex, covering both complex single-agent and two multi-agent domains against reasonable baseline models.
- The paper is very nicely written and well organized. The logical structure of motivation, related work, background, analytic study of the frameworks in combination with the metric and the presentation of the SIPO algorithm and its evaluation is easy to follow and intuitively explained. Small examples to illustrate the argumentation help lighten an otherwise densely formulated reasoning. I also find the visualizations to be nicely done.
- The author’s SIPO approach seems to produce some very diverse strategies, which also appear intuitively interpretable. While diversity in itself is very important for exploration and an important Problem to solve for RL in general, I find the convergence towards interpretable strategies to be very intriguing. The paper is not only nicely understandable and informative on the matter of diversity related approaches, the evaluation is rather strong and the supplementary material has enough content for another paper by itself. I have no real complaints about this work, very nicely done.

**Weaknesses:**

- Besides providing a reasonably complex evaluation, there could have been more than 5 seeds used for the evaluation.
- I would also like to see at least some classic total-reward metrics / comparisons for these domains, as the metrics chosen here to highlight the performances of the SIPO algorithm (pairwise distances, est. state entropy, agent position distributions, nr. distinct strategies) seem a bit cherry-picked to argue in favor of diversity only. While there are some relevant plots in the Appendix (and some of the toy-examples reason with rewards), I would encourage to showcase at last one such classic performance comparison against the baselines in the main-paper itself.


**Questions:**

- In 4.2 it is mentioned that PBT only converges when the exploration is faint. Is this claim shown somewhere or is this based on empirical evidence?
- Could you provide some motivation on why you are casting $D_s(\pi_i, \pi_j^*)$ as intrinsic reward, and why RBF Kernel and the Wasserstein Distance were chosen for the diversity? Why does L_2 WSD provide ‘stronger discriminative power’(l263)?
- Increasingly more MARL approaches start to require a DEC-POMDP for independent / localized actions without a fully observable state. How would you judge the SIPO’s transferability to such a formalization?

**Limitations:**

There are two larger limitations mentioned, for once the assumption of continuity for convergence and the access to an object-centric state-representation. Both are openly disclosed and discussed in the paper. ‘The acceleration of ITR remains an open challenge’ is also a valid outlook, that should be addressed in the future.

---

> ### Author Rebuttal · Authors · 2023-08-10
>
> + I would also like to see at least some classic total-reward metrics / comparisons
>
> Appendix B.5 elaborates on the detailed returns accomplished by SIPO. We present the diversity score and average rewards achieved by all algorithms below. These numerical values are averaged across the entire population for a clear comparison:
>
> | humanoid	|SIPO-RBF 	| SIPO-WD 	| DIPG 	| RSPO | SMERL 	| DvD | PPO |
> |  ----  | ----  |----  |----  |----  |----  |----  |----  |
> | reward 	|	3508 	| 	3763 	| 5191	 | 1455   |4253 | 4498 | 5299 |
> | diversity 	| 	0.53	| 	0.71 	| 0.12 	  | 0.53   | 0.01 | 0.40 | -   	|
> | SMAC 2m1z  |		|		|	|	|	|       |	        |
> | win rate %   | 	100 	| 	100 	| 100	| 100 	| 100 |100 | 100 |
> | diversity (1e-3) | 	38 	| 	36 	| 32	| 32	| 28  | 30   | -     |
> | SMAC 2c64zg |		|		|	|	|	|    |	|
> | win rate % 	| 	99 	| 	93 	| 99	| 85 	| 100 |100 | 100 |
> | diversity (1e-3) | 	72	| 	56 	| 70	| 56	| 42    | 57 | -     |
> | GRF 3v1 (first 4) |		|		|	|	|	|    |	|
> | win rate % 	| 	93	| 	 82	| 93	|  94	|   91     |   83   |    92   |
> | diversity 	|  	3.0	|  	3.0	| 2.7	| 2.0    |   1.3      |   3.0   | 2.7     |
> | GRF CA 	|		|		|	|	|	|       |		|
> | win rate % 	| 	70	| 	41	| 46	|  76	|    45    |   -    |   50    |
> | diversity 	|  	3.3	|  	3.0	| 2.3	| 2.0	|   1.3      |   -   | 1.7     |
> | GRF Corner 	|		|		|	|	|	|    |	|
> | win rate % 	| 	72	| 	56	| 75	|  23	|    67    |    -   |   71    |
> | diversity 	|  	2.7	|  	3.0	| 1.7	| 1.6	|   1.0      |   -   | 2.0     |
>
> The tabulated data above highlights the varying trade-offs between task performance and diversity exhibited by different algorithms. It is noteworthy that SIPO, in particular, displays an adeptness at training a notably more diverse population while upholding a reasonably moderate level of task performance.
>
> + In 4.2 it is mentioned that PBT only converges when the exploration is faint. Is this claim shown somewhere or is this based on empirical evidence?
>
> This conclusion can be drawn from empirical evidence presented in Table 2. While PBT may occasionally succeed in learning diverse policies, this outcome is contingent upon initializations that prompt the agent to explore all landmarks. Nevertheless, such instances of success are outweighed by cases of failure, leading to notable variance in PBT outcomes.
>
> + why you are casting the diversity measure as intrinsic reward
>
> The objective in Equation (7) is not differentiable w.r.t. \pi. This is because it depends on the states traversed by \pi, rather than \pi’s output. Consequently, conventional gradient-based methods are unsuitable for its optimization. Nevertheless, we are able to represent it as the cumulative sum of intrinsic rewards, specifically the intrinsic return. This allows us to leverage policy gradient techniques for optimization.
>
> + why RBF Kernel and the Wasserstein Distance were chosen for the diversity? Why does L_2 WSD provide ‘stronger discriminative power’(l263)?
>
> We first choose the RBF kernel because it is the simplest and the most widely adopted kernel function in the ML community. Regarding the choice of the Wasserstein distance, also referred to as the earth-moving distance, it holds a distinct advantage due to its interpretation within optimal transport theory[1,2]. Unlike distances that rely solely on specific summary statistics such as means, the Wasserstein distance can effectively quantify shifts in state distributions and remains robust in the presence of outliers[2].
>
> [1] Arjovsky, M., Chintala, S., & Bottou, L. (2017, July). Wasserstein generative adversarial networks. In International conference on machine learning (pp. 214-223). PMLR.
>
> [2] Villani, C. (2009). Optimal transport: old and new (Vol. 338, p. 23). Berlin: springer.
>
> + Increasingly more MARL approaches start to require a DEC-POMDP for independent / localized actions without a fully observable state. How would you judge the SIPO’s transferability to such a formalization?
>
> Experimental outcomes in SMAC and GRF confirm the favorable performance of SIPO in Dec-POMDPs. Like most popular MARL algorithms, we assume accessible global states only during training, i.e., centralized training with decentralized execution (CTDE).
>
> In the fully decentralized training setting, discovering diverse policies requires the fundamental improvement of existing MARL algorithms. While this is not the main focus of our paper, we are making efforts to address this problem in future works.

---

> > ### Comment · Reviewer_tun3 · 2023-08-18
> > **Thank you for the rebuttal**
> >
> > Thank you to the authors for this honest reply. While Apx.B5 does in fact show some reward details, the averaged scores on this provided table much more clearly show that there is in fact quite a tradeoff between diversity and performance. I hope this will be communicated more upfront in the CR. However, I do personally agree with the authors on the importance of diversity and interpretability of RL-policies, even though performance is not quite on par. I am still quite in favor of publishing this work due to its excellent readability and the progress on human interpretable results. I would welcome it if it code would be open-sourced as an helpful tool for the community.

---

### Author Rebuttal · Authors · 2023-08-10

We express our sincere gratitude to the reviewer for their meticulous examination and thoughtful feedback on our manuscript. We response to each reviewer's question in the corresponding channels. Please feel free to drop a message if you have additional concerns.

We acknowledge a typo in the figures of our paper. "IL" should be "ITR". We promise to fix this in our next revision.

The attached pdf contains the heatmap of DIPG in SMAC, in response to reviewer hytR.

---

### Decision · Program_Chairs · 2023-09-21

**Decision:**

Accept (poster)

**Comment:**

The paper had mixed reviews, with most reviewers weakly accepting it following discussion with the authors, one reviewer accepting it and one reviewer rejecting it. I have read the paper, the rebuttal and discussed it with the reviewers and I feel that there is still some unclarity w.r.t to Equation 5 (which is the main contribution of the paper) that must be further discussed. In the rebuttal, the authors explain how Equation 5 can be used to encapsulate previous diversity objectives, including Successor Features based diversity. It allows the authors to introduce a loss function between states inside the expectation over the state occupancy, while applying it after the expectation will result in diversity based on SFs. Introducing a non linear function before taking an expectation might bring various advantages. In particular, it is often the case that when averaging the features over the occupancy, to achieve successor features, some information is lost due to averaging. This is a great novelty by the authors, that leads to interesting results. I would like to see that part in the paper clearly discussed w.r.t to related work. I also agree with the reviewers that there is currently a gap in understanding this new objective theoretically. However, given the empirical contributions of the paper, I think that the evidence is clear that this is an interesting idea and that other people might want to study it. I would like to see some discussion on future work on this front, what is missing and why. The authors gave some explanations on that in the rebuttal and they should be included in the paper. I encourage the authors to implement the feedback from the reviews, as well as the new results posted during the rebuttal when making their final version. I will follow closely to see how these changes were implemented. I recommend accepting the paper.